# A Twitter-Lived Red Tide Crisis on Chiloé Island, Chile: What Can Be Obtained for Social-Ecological Research through Social Media Analysis?

**Aldo Mascareño [1,2,\*], Pablo A. Henríquez [3], Marco Billi [2,4] and Gonzalo A. Ruz [5,6]** 

1    Centro de Estudios Públicos, Monseñor Sótero Sanz 162, Providencia, Santiago 7500011, Chile
2    Escuela de Gobierno, Universidad Adolfo Ibáñez, Diagonal Las Torres 2640, Peñalolén, Santiago 7941169, Chile; marco.dg.billi@ug.uchile.cl
3    Facultad de Economía y Empresas, Universidad Diego Portales, Av. Sta. Clara 797, Huechuraba, Santiago 8581169, Chile; pablo.henriquez@udp.cl
4    Center for Climate and Resilience Research (CR)2, Universidad de Chile, Blanco Encalada 2002, Santiago 8370449, Chile
5    Facultad de Ingeniería y Ciencias, Universidad Adolfo Ibáñez, Diagonal Las Torres 2640, Peñalolén, Santiago 7941169, Chile; gonzalo.ruz@uai.cl
6    Center of Applied Ecology and Sustainability (CAPES), Santiago 8331150, Chile
*    Correspondence: amascareno@cepchile.cl

**Abstract:** Considering traditional research on social-ecological crises, new social media analysis, particularly Twitter data, contributes with supplementary exploration techniques. In this article, we argue that a social media approach to social-ecological crises can offer an actor-centered meaningful perspective on social facts, a depiction of the general dynamics of meaning making that takes place among actors, and a systemic view of actors' communication before, during and after the crisis. On the basis of a multi-technique approach to Twitter data (TF-IDF, hierarchical clustering, egocentric networks and principal component analysis) applied to a red tide crisis on Chiloé Island, Chile, in 2016, the most significant red tide in South America ever, we offer a view on the boundaries and dynamics of meaning making in a social-ecological crisis. We conclude that this dynamics shows a permanent reflexive work on elucidating the causes and effects of the crisis that develops according to actors' commitments, the sequence of events, and political conveniences. In this vein, social media analysis does not replace good qualitative research, it rather opens up supplementary possibilities for capturing meanings from the past that cannot be retrieved otherwise. This is particularly relevant for studying social-ecological crises and supporting collective learning processes that point towards increased resilience capacities and more sustainable trajectories in affected communities.

**Keywords:** social-ecological crisis; social media analysis; meaning-making; learning processes; Twitter data; red tide; Chiloé Island

## 1. Introduction

Global climate and environmental changes increase in depth and extension in different regions of the world [1] and bring forth the need for a deep and urgent transformation of our society as a whole and the way social analysis deals with the many levels those general, eventually sudden changes involve.

From a scientific point of view, climate and environmental changes are translocal, if not global processes [2–7]. They combine different natural levels of species-landscape interaction as well as different layers of reality, namely, natural, social and technological, that relate with each other in complex ways [8–11]. From the point of view of social actors, however, the network of repercussions and exchanges arising from social-ecological events are regularly a locally-lived phenomenon. Certainly, some actors may be more aware of the fact that what they locally experience is related to other local experiences in different geographical spaces, and also to transnational structures and networks with a global scope. But for most of them who are not scientists or corporate actors with vested interests, the reality and meaning of a natural or social-natural event come defined by the concrete social-spatial position in which the event is experienced and interpreted [12,13]. To put it in the traditional terms of social sciences, structure and agency represent different points of entrance into social analysis [14].

In line with this difference, social sciences have developed special methods for capturing structural realities (mostly viewed as aggregate of individual preferences in quantitative research) and actors' experiences (mostly regarded as phenomenologically lived-events in qualitative research). With the rise of social media and big data analysis, however, new opportunities for social research arise, which challenge accustomed paradigms in the social sciences. The difference between qualitative and quantitative methods becomes rather blurred [15] because of several reasons: social media cover a variety of communicative flows (interpersonal communication, official communication, citizen sensing, interorganizational communication), they work in real time, and can capture the big picture of a critical situation and combine it with information on actionable insights [16]. It is, thus, our contention that by applying suitable methods to social media information, it becomes possible to associate the structural approach to social-ecological crises –namely, the behavioral patterns and complexly interrelated semantic networks– with the more phenomenological actor-centered approach—i.e., shared meanings, common experiences, and learning processes oriented to reinforce community resilience and the transformation towards sustainability coming from different social actors. This attempt of constructing a mixed approach to social-ecological systems (SES) by addressing structural realities and actors' experiences through social media analysis can contribute to the discussions on the role of the social in SES literature. Human impacts on the environment neither come from isolated individual decisions nor from their simple aggregation, but from socially constructed constellations of meaning wherein values, knowledge, social diversity and power relations play a fundamental role. Observing how these meaningful constellations work and how they motivate resilience practices and learning processes in crisis situations is thus crucial for a better understanding of the social-ecological nexus [17–25].

Additionally, this effort can contribute to the emerging scholarship in the realm of neogeography, volunteered geographical information, and participatory geographic information systems. This scholarship signals the opportunities associated to crowd-sourced information and other sources of big data for the analysis of a variety of social phenomena [26], including crises [27,28], environmental degradation [29], inequalities [30], and disaster consequences and response [31–33]. Neogeography has also highlighted how the growing availability and accessibility of user-generated information transforms the comprehension of actors, practices and contents of social-geographical analysis and place-making [34,35], thereby opening new avenues for citizen engagement in knowledge production [36,37] and generating new challenges in terms of quality assurance [38,39] and critical analysis [40].

Taking this framework into consideration, in this article we aim to identify what can be obtained for social-ecological research through social media analysis, particularly Twitter analysis, of a crisis situation. Our case-study is the social-ecological red tide crisis on Chiloé Island, Southern Chile, in 2016, the most extended and intense red tide in South America ever recorded [41]. In a previously published article [42], the authors have analyzed this event in detail mostly under an actor-centered qualitative approach with a few elements of social media research. In the meantime, other researchers have applied similar methods to

explore this case [43–45], while others conducted a more structure-centered research, whether referring to quantitative ecological data [41,46] or to communicative consequences [47,48]. Considering this, we argue that the case of the 2016 red tide crisis on Chiloé Island, is a well-suited case for identifying the novel knowledge that can be gained for social-ecological research when conducting social media analysis with a multi-method approach. From this exploration, we conclude that social media and particularly Twitter analysis, when performed through multiple techniques and interdisciplinarily conducted, offers (a) a structural systemic view of the semantic boundaries and networks of a social-ecological crisis, (b) an involvement with the meaningful, phenomenological aspects of these events, and (c) a depiction of the processes of meaning making that take place among actors and that can be a source of transformation efforts towards a more sustainable future.

In order to accomplish this, we begin with a literature review of the intersection between social media analysis and social-ecological crises aimed at identifying the different research lines on the topic and positioning our research. Secondly, we offer a characterization of the 2016 red tide crisis on Chiloé Island, Chile, with a view on the different methods applied by recent research and the results obtained in each case. Next, we describe our multi-method approach, and then we present our results in dialog with the qualitative and quantitative research conducted on the same case and on social-ecological crises in general, and discuss them regarding the aims of our study. Finally, we draw the conclusions from our work.

*Literature Review: Social Media and Social-Ecological Crises*

Social media are increasingly recognized as a promising tool to gather and analyze data which would otherwise be inaccessible or excessively costly and time consuming [49,50]. During the last years, particularly the scientific literature linking Twitter and other social media with social-ecological crises and emergencies has grown [51]. Most scholarship in this field was published from 2013 onward. The most relevant disciplinary approaches are—in a descending order—communication research, information and computer science, business and management, social sciences (e.g., psychology, sociology, geography) and public health.

Four main strands of literature can be distinguished. The first and most established one refers to the use of Twitter (and social media) as a tool for crisis communication and reputation management in private corporations. The literature either highlights the potential relevance of these tools, reports on case studies or examines the differential effectiveness of particular practices and strategies [52–56].

Related to the above, a second strand of literature examines the use of Twitter by public agencies. Usually, this scholarship focuses on one or few case studies analyzing the growing relevance of these media as tools for supporting vulnerability assessment [57] and facilitating community resilience [58] as well as integrated disaster responses [59] and territorial planning [60]. While much literature adopts a positive stance towards the use of these methods, often offering recommendations for public agents, other contributions adopt a more critical tone, focusing on how agencies select information strategically, on the politics behind the use of these media, its limits and underlying risks [61–63].

A third strand of literature discusses the potential of Twitter and other social media as sources of information to reduce risks, identify and prevent crisis situations and put in place early warning systems. This scholarship debates different advanced methods and techniques for Twitter mining, labelling and automated screening, as well as the potential usefulness of crossing social media data with other information sources such as geospatial data [64–69]. This literature shows the relevance of Twitter information during disasters so that authorities can make better decisions [70] and also highlights Twitter importance to create awareness and minimize possibles damages [71–73].

A fourth and last strand of literature offers insights on the dynamics and potential performance of Twitter communication, investigating a broad variety of questions, including:

- What motivates the behavior of Twitter users, e.g., how do they deal with uncertainty in Twitter communication, what drives twitting and re-twitting, the emergence and role of influential social media users in disaster situations [74–76];
- How social media become the space for controversies and 'issue arenas' [77] and the influence they have on crisis situations [78], e.g., promoting political upheavals and situations of social change in regions affected by crises or natural disasters [79–81];
- How Twitter and other social media interrelate with traditional media, e.g., how well they fare at different functions, how they overlap and feed each other [82,83] and how routine social media communications and networks are affected by disaster situations [84];
- How Twitter and social networks spread information (and misinformation) about crises, e.g., modeling information diffusion in Twitter networks, assessing its ability to reinforce risk communication, understanding the birth and reproduction of fake news and hoaxes, among others [85–90];
- Finally, how does Twitter drive sense-making about crisis, and what kind of contents, framing narratives and sentiments are dominant in different contexts, groups of actors, and stages of a crisis.

This last research trajectory is the most relevant for the study proposed in this article. Evidence from different types of crisis situations suggests two overarching trends in Twitter activity: (a) the emergence of plural and often competing sets of meaningful interpretations of the crisis (narratives) containing different farmings, semantics and sentiments elicited by diverging actors' experiences, which give shape to different 'publics' or interpretative communities that either pre-exist or emerge as a result of the crisis itself [78,91–94]; and (b) the tendency of crisis communication to evolve from an initial acute stage characterized by the breakdown of accustomed structures of meaning, a shared feeling of uncertainty, and the cacophonic, noisy and irreflexive spreading of information, to a more evaluative and retrospective exercise of sense-giving from which a defined interpretation emerge [95–99].

Noticeably, while these patterns have been studied with respect to both social-political controversies and natural disasters, little understanding exists so far about how they manifest in social-ecological crisis situations, where the immediacy of the ecological trigger (the 2016 red tide in Chiloé) inlays into a pre-existing situation of latent and self-induced criticality (historical discontent and transformations in the Chiloé case study). Understanding sense-making or meaning-making can be key to assess whether social-ecological crises activate learning processes in public opinion, thereby pushing it to embrace transformative stances that might pave the way towards more sustainable trajectories.

In this vein, social-ecological systems (SES) literature is a fertile field of research. The SES approach investigates the complex adaptive system dynamics linking human and natural systems [100], explores their trajectories and interdependencies at multiple scales [101,102] as well as pathways for the sustainable governance of these systems [103,104]. In recent years, increasing attention has been devoted to the role played in social-ecological systems sustainability by mismatches of scale between social-ecological phenomena and institutional arrangements [105], time-delayed ecological feedbacks on biodiversity upon human activities [106], the relevance of values, knowledge, social diversity and power relations in the processes of meaning-making leading to resilience practices, learning, and sustainability [17–24], and the complex interactions between interlocked social-ecological systems [107], events which may have been playing a salient role in the 2016 Chiloé red tide crisis.

Fisheries and fishing ecosystems, in particular, have been abundantly explored within social-ecological research. Fishing resources are a classical example of a common-pool resource [108], and thus prone to over-exploitation and under-caring. This has led to ample interest in exploring the possibility of

setting up sustainable fishing communities, particularly through the use of polycentric frameworks and principles [109,110], and has also led to investigate the resilience of fishing communities in the face of internal and external stresses [111–113].

Within this context, Chiloé is a widely employed case study on the sustainability of fishing ecosystems, especially in association with the sustainability of salmon and shellfish aquaculture and their environmental and social impacts on the island and its population [114–118]. The 2016 events have contributed to boosting this literature, with new contributions multiplying in recent years, covering long-term socio-cultural and social-ecological trends and transformations contributing to explain the crisis [44,119,120], the specific influence played by the salmon dumping on the crisis [46]; local perceptions on the crisis and its relationship with climate and environmental change [121] and the different frames employed in media coverage of the crisis [47,122]. Noticeably, only one of these studies [48] makes use of social media analyses, and as we observe in detail in the next section, it is focused on the limited use of these platforms for concrete political actions than on the evolution of social media communications as a way to understand the underlying dynamics of the crisis.

## 2. Background: The 2016 Red Tide Crisis on Chiloé Island

Chiloé Island is located between latitudes $41°4'$ and $43°2'$ S in southern Chile (Figure 1). According to the last available data [123], the population of the island amounts to 180,185 inhabitants, divided into 10 municipalities. Until 1980, Chiloé's economy depended on subsistence-oriented agriculture, artisanal fishing and woodcutting [114]. From the 1980s onwards, the island became a center of the world salmon industry. Currently, it is the second global producer after Norway [124].

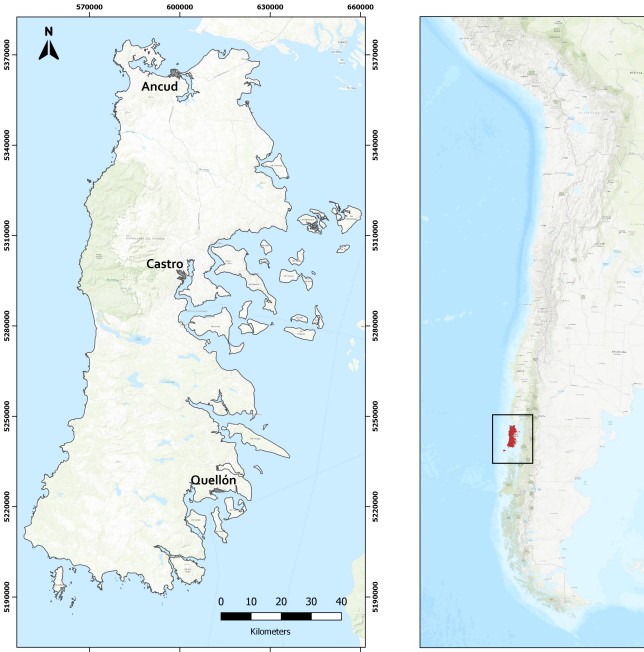

**Figure 1.** Chiloé Island, Chile.

As a consequence of the intensive aquaculture, several crises have affected the region in the last decades [125], being the 2016 red tide crisis the last one. On February 2016 the most significant algal bloom in the history of Chiloé Island was declared. The affected area comprehended the interior and exterior

coast of the Island and the Reloncaví Gulf. The following factual events constitute the basic structure of the crisis:

- February 2016: First algal bloom (first pulse) producing a mortality of 25 million fish (39,942 t) by the second week of March 2016;
- 10 to 26 March 2016: Dumping of 4600 t of dead fish 75 nautical miles off the northwest coast of Chiloé Island;
- Third week of March 2016: Second algal bloom (second pulse);
- 19 March 2016: Ban on fishing and shellfish extraction issued by the Chilean Ministry of Health covering nearly 2000 km of coast;
- 25 April 2016: Massive stranding of shellfish on in Chiloé's coasts;
- 29 April 2016: Chiloé is declared 'catastrophe zone' by the Chilean government;
- 2 to 19 May 2016: Massive demonstrations of Chiloé's inhabitants against government's measures and the salmon industry that took over the island;
- Late May 2016: End of demonstrations thanks to a controversial agreement with the Chilean government [42,44].

As stated in the Introduction, this crisis has been mostly studied under an agent-centered qualitative approach [42–45], while other authors have conducted a structure-centered research based on quantitative ecological data [41,46] or on communication analysis [47,48]. Table 1 summarize these research approaches on the 2016 red tide crisis on Chiloé Island and their main conclusions.

**Table 1.** Research on the 2016 red tide crisis on Chiloé Island.

| Reference | Year | Keywords | Methods | Conclusions |
|---|---|---|---|---|
| Ref. [47] | 2018 | Chiloé; socio-environmental; contentious politics; framing; red tide | Framing method (communication sciences) and discourse analysis of online press | Nature is responsible for the crisis; origins of the crisis are beyond the salmon industry. The conflict is due to disagreements regarding monetary compensations. |
| Ref. [42] | 2018 | Chiloé Island; controversy; governance; regime shift; resilience; salmon aquaculture | In-depth interviews, content analysis of press articles and archives, and semantic analysis of Twitter data (5104 tweets from 2013 to 2017) | Chiloé's social-ecological system (SES) is an unstable regime prone to sudden shifts. Controversies manipulate the epistemic construction of SES. This requires to be managed through non-authoritative forms of governance |
| Ref. [48] | 2018 | Politics; technology; popular mobilization; social media; Internet; Twitter | Content analysis on Twitter data (409 tweets on April 2016) | Findings suggest that social media are mostly used for sharing information rather than promoting or instructing Twitter users to perform concrete political actions |
| Ref. [44] | 2019 | Chiloé; conflict; islandness; islands; salmon farming; social mobilization | Qualitative analysis based on in-depth interviews | Islandness in Chiloé Island is a political stance in opposition to the state. It expresses the power asymmetries between islanders and the Chilean state |
| Ref. [126] | 2019 | Social-ecological systems; Latin America; complexity; Chile; environmental governance; Chiloé | Social-ecological survey | Several factors work against the adaptation of Chiloé's ecosystem services, such as dominance of governmental institutions, low participation, lack of collective learning, market influences, centralized decision-making, among others |
| Ref. [46] | 2020 | Harmful algal blooms; aquaculture, pollution control; ocean transport; ecological crisis; risk management | Oceanic modeling (Mercator model) and transport simulations (Lagrangian transport code Ariane) | The analyses show consistently that the dumping of dead fish into the sea could well have played a fueling role for the second pulse of the red tide |
| Ref. [43] | 2020 | Book chapter; no keywords | Content analysis of three Facebook fan pages in Chiloé between 2016 and 2018 (n = 2937) | Social actors continue posting on the crisis long after the phenomenon passed. Digital media continue distributing news on the topic long after legacy media lost interests |
| Ref. [45] | 2020 | Chiloé; identities; salmon industry; territory | Content analysis of semi-structured interviews | Tensions between local identities limit the projection of the island identity because of a non-accomplished self-recognition |

As seen, most of the research on the 2016 Chiloé red tide crisis follows qualitative criteria. In-depth interviews are the most preferred source of data, while content analysis is the most frequently selected technique. Three studies [42,43,48] resort to social media analysis in their design, while one of them [126] uses survey data for assessing Chiloé's social-ecological services, and another one [46] applies modeling and simulations for its purposes, being this an oceanographical analysis of the triggering role that the dumping of dead salmons into the sea played in the 2016 red tide crisis.

By drawing on the set of multiple techniques and procedures presented in the next section, we aim to offer an understanding of the social-ecological red tide crisis that integrates the meaningful components of qualitative research with a systemic general view on the network boundaries and its most characteristic behavioral patterns.

## 3. Data and Methods

This research uses two independent datasets. The first dataset was obtained from Google Trends to confirm the geospatial interest on the red tide. The second dataset was obtained from Twitter to analyze semantic contents in three periods: before, during and after the crisis.

### 3.1. Google Trends (GT)

Google Trends (https://trends.google.com/trends/) is a service that outputs the time series data of search intensity to show the extent to which a particular keyword is searched for in a specified period and location [127]. Search intensity is an index of search volume adjusted to the number of Google users in a given geographical area. This value ranges from 0–100, where the value of 100 indicates the peak of popularity (100% of popularity in a given period and location) and 0 complete disinterest (0%). Google data dates back to 2004 and can be provided as a daily or monthly data.

GT have been extensively used in various research domains, such as epidemic forecasting, financial analysis, marketing, tourism and general research exploring public opinion [128]. GT may qualify analyzed phrases as "search term" or "topic". Search terms are literally typed words, while topics may be proposed by GT when the tool recognizes phrases related to popular queries.

The search intensity data are obtained from Google Trends. We specify the region as CL (Chile) and use two terms "marea roja" (red tide) and "Chiloé" (Chiloé) as search keywords. We collected the Google data from 1 January 2004 to date of collection (1 February 2020). The data were obtained using R 3.6.1 gtrendsR package version 1.4.4 to search the time series.

### 3.2. Twitter Data

We used the public timeline API method provided by Twitter to collect information about the non-protected users who have set a custom user icon in real time. In our case, users were placed into four main groups that predominantly relate to four social positions: technical agents (Actors 1), non-governmental organizations (Actors 2), fishermen social organizations (Actors 3), and political authorities (Actors 4). Details of the user, such as IDs, screen name, and date were extracted. We used a Python library called Tweepy to connect to Twitter Streaming API and download the data. The dataset is a random sample of 27,935 tweets (including retweets) captured over the period 2013–2017 in order to differentiate three periods in our analysis: before the crisis (from 2013 to February 2016), during the crisis (from March to May 2016) and after the crisis (from June 2016 to January 2017). This sample includes tweets from 11 unique users (see Table 2).

**Table 2.** Actors and Twitter accounts.

| Actors | Twitter Accounts | Number of Tweets |
|---|---|---|
| Actors 1: Technical agents (Fishing National Service, Institute for the Promotion of Fishing, Chilean Army, Association of Salmon Farmers) | @sernapesca, @ifop_periodista, @Armada_Chile, @SalmonChileAG | 9794 |
| Actors 2: Non-governmental organizations (Greenpeace, Crea Foundation) | @GreenpaceCL, @fundacioncrea | 4781 |
| Actors 3: Fishing organizations (National Association of Artisanal Fishers, Federation of Fishermen) | @ConapachChile, @fetrapes | 3673 |
| Actors 4: Political authorities (Ministry of Economy, Subsecretary of Fishing, Intendancy Los Lagos) | @meconomia, @subpescaCl, @intendencialos1 | 9687 |
| Total | | 27,935 |

### 3.3. Pre-Processing

Raw data is highly susceptible to inconsistency and redundancy. Preprocessing of tweets is necessary and include the following points:

- Remove all URLs (e.g., www.xyz.com), hashtags (e.g., #topic), targets (@username);
- Remove all punctuations, symbols, numbers;
- Correct the spellings; sequence of repeated characters is to be handled;
- Remove stop words;
- Remove non-Spanish tweets.

Then, to analyze the actors' tweet in relation to the red tide event, we selected only those tweets containing the keywords in Table 3. The final corpus for the four groups of actors amounts to 5104 tweets. This analysis was conducted using the open-source R software environment for statistical computing.

**Table 3.** Keywords for the selection of tweets.

| Original words in Spanish | Translation |
|---|---|
| Crisis, desastre, alga, nocivo, paralización, protesta, floración, verter, vertedero, cambio, climático, ley, pesca, artesanal, muerto, salmón, calentamiento, niño, marea, roja, eutrofización, vertimiento, varazón, molusco, marisco, mariscador, petitorio, sacrificio, deliberado, zona, catástrofe, bono, buzo, mortandad, toxico | Crisis, disaster, algae, harmful, paralyzing, protest, bloom, dump, dumping zone, change, climatic, law, fishing, artisanal, dead, salmon, warming, niño, tide, red, eutrophication, dumping, stranding, mollusc, shellfish, shell fisherman, request, sacrifice, deliberate, zone, catastrophe, bonus, diver, mortality, toxic |

### 3.4. Term Frequency-Inverse Document Frequency (TF-IDF)

Term frequency-inverse document frequency (TF-IDF) is a numerical statistic that demonstrates how important a word is to a corpus -a very popular research method in the field of natural language processing (NLP) [129] which we used in the context of the red tide. TF-IDF method determines the relative frequency of words in a specific document through an inverse proportion of the word over the entire document corpus. The method uses two elements: Term frequency (TF) of term $i$ in document $j$ and inverse document frequency (IDF) of term $i$. TF-IDF can be calculated as [130]:

$$a_{ij} = tf_{ij}idf_i = tf_{ij} \times log_2\frac{N}{df_i},$$

(1)

where $a_{ij}$ is the weight of term $i$ in a document $j$, $N$ is the number of documents in the collection, $tf_{ij}$ is the term frequency of the term $i$ in a document $j$ and $df_i$ is the document frequency of the term $i$ in the collection.

### 3.5. Hierarchical Clustering

Among the many clustering algorithms, we used the agglomerative hierarchical clustering approach to identify clusters of words based on the TF-IDF matrix, where each row represents the weights of a word for different documents (tweets). The agglomerative version of hierarchical clustering assumes that each data point is an initial individual cluster. Then, in each iteration, the two closest (under some criterion) clusters are merged. One of the advantages of using agglomerative hierarchical clustering is that it is not required to specify the number of clusters manually beforehand, like in k-means. The clusters are naturally identified through the resulting dendrogram representation. In this work, we used Ward's method, which minimizes the total within-cluster variance, as the criterion for merging clusters (words) during the clustering process.

### 3.6. Egocentric Networks

In this study, we use the method called egocentric networks to identify significant subnetworks that are associated with the red tide. In particular, an ego-network is the part of a network that involves a specific node we are focusing on, which we call ego. The network consists of a neighborhood, with all the nodes connected to the ego, at a particular path length. We create subnetworks were the nodes are words from the corpus. Two nodes (words) are connected with an edge if and only if there is a bigram that contains them. Each edge has a weight equal to the number of bigrams (frequency) of this kind generated from the analyzed dataset.

## 4. Results and Discussion

The fundamental aim of our analysis is to identify and relate two different layers of social exploration in social-ecological crises, namely, the actor-centered meaningful components of the crisis' experience in the 2016 red tide event in Chiloé Island as well as the general dynamics of actor's behavior and communications.

To begin with, we present two time series of the most important concepts of our research: Chiloé and red tide. Data has been recovered from Google Search between 2004 and 2020. Then, we offer a more detailed time series of Twitter's utterances of the four actors of our research (technical agencies, NGOs, artisanal fishermen, and political authorities) between 2013 and 2017, so that we can capture the periods before, during and after the crisis with our selected keywords (Table 3). In a second step, we identify the meaningful semantic components from actors' point of view; we do this by analyzing word clouds by actors before, during and after the crisis combined with the examination of egocentric networks for different actors and periods. In a third step, we move to a higher level of observation by examining the semantic clustering of the whole corpus in the three periods. Topics and tweets are analyzed to reconstruct the external and internal semantic boundaries of the different phases of the 2016 red tide crisis. Finally, in a fourth step, we present a principal component analysis of actors' utterances to show their topological distribution and the form they semantically overlap and differentiate. In Table 4 we summarize our main findings which we describe in detail below.

**Table 4.** Summary of techniques and findings.

| Technique | Focus | Analysis |
|---|---|---|
| Time series | Time series present the period under consideration with two main words for an introductory characterization and offers a general view on the frequency of utterances by actors. | Chiloé is regularly viewed as a touristic zone. In 2016 the island was the center of attention because of the red tide. The group of actors we follow for the analysis are active on Twitter long before the crisis. |
| Word clouds | The technique identifies the meaningful semantic components from actors' point of view. It offers a detailed view on both the central and peripheral concerns of different actors before, during and after the crisis. It also allows to compare semantic similarities and differences between actors and periods. | 'Fishing' and derivatives (artisanal fishing, fishermen, fishing regulations, fishing industry) are central concerns of Chiloé inhabitants. As such, the 2016 red tide becomes relevant for technical agencies and NGOs, while fishermen experience the crisis as a problem of employment. NGOs preserve the memory of the crisis after the event. |
| Egocentric networks | These networks offer a detailed view on the relevant meaningful components of the considered periods. The analysis is conducted by periods and actors with different focal terms concepts. | Artisanal fishing and artisanal fishermen appear as central sources of meaning-making in Chiloé Island. Semantic reflections of critical events (stranding of shellfish and dumping of salmons) are produced during the crisis. |
| Clustering | This technique identifies proximity of words or tweets in the corpus. We apply this to recognize internal and external semantic boundaries in actors' communication. These are expressed in words' and tweets' aggregation. | Before the crisis, three thematic groups of tweets can be distinguished. The crisis produces a reduction in the variety of themes, which cluster more tightly together; while after the crisis, the themes multiply. A similar dynamics arises when clustering is applied to tweets instead of topics. |
| Principal component analysis | The technique displays actors' tweets distributed into a two-dimensional space, thereby allowing to identify distance and proximity between centroids and revealing the topological borders of both actors' communications and the complete semantic area. | Before the crisis an important group of fishermen's utterances do not intersect with communications of other actors. During and after the crisis, NGOs' communications become closer to those of fishermen, yet with different emphasis. |

### 4.1. Time Series of Keywords and Actors' Utterances

We present two time series offering a general view on our subject (the 2016 red tide in Chiloé Island) on the one hand, and on the corpus we use for analysis, on the other. In Figures 2 and 3 we observe the behavior of two basic words: "Chiloé" and "red tide", from 2013 to 2017.

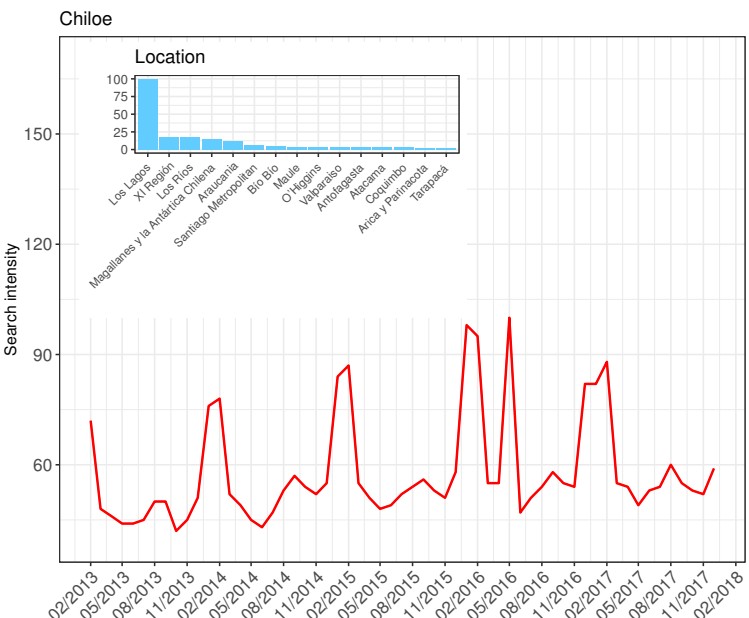

**Figure 2.** Time series of the keyword "Chiloé", 2013–2017.

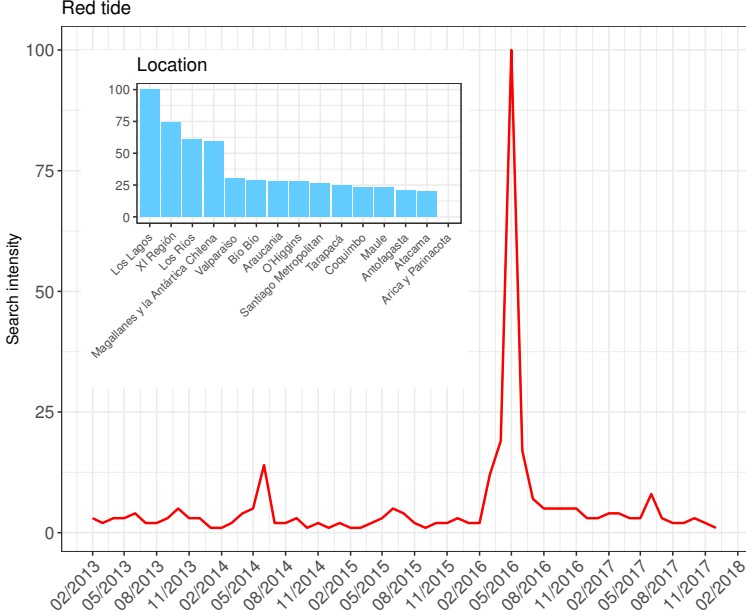

**Figure 3.** Time series of the keyword "red tide", 2013–2017.

In Figure 2 we notice a clear upward trend in Chiloé searches that begins before the crisis (2009), with cyclical surges before the summer season (probably related to tourism). In 2016, we observe that a non-seasonal surge appears at the moment of the crisis, thereby showing that the island captured wider public attention. However, the surge is short-lived, and the searches quickly go back to their cyclical form, with non-appreciable changes in the general trends. Figure 3 explains the short-lived surge in 2016. As seen, the concept of "red tide" presents a massive spike in searches during the crisis period, but with

no evident aftereffects. The general attention drops after the crisis as rapidly as it had surged, and remains mainly as a local phenomenon, as argued by [47].

Going deeper into our analysis, we can observe in Figure 4 a general framework of the research period in a time series of actors' utterances in Twitter data from 2013 to 2017.

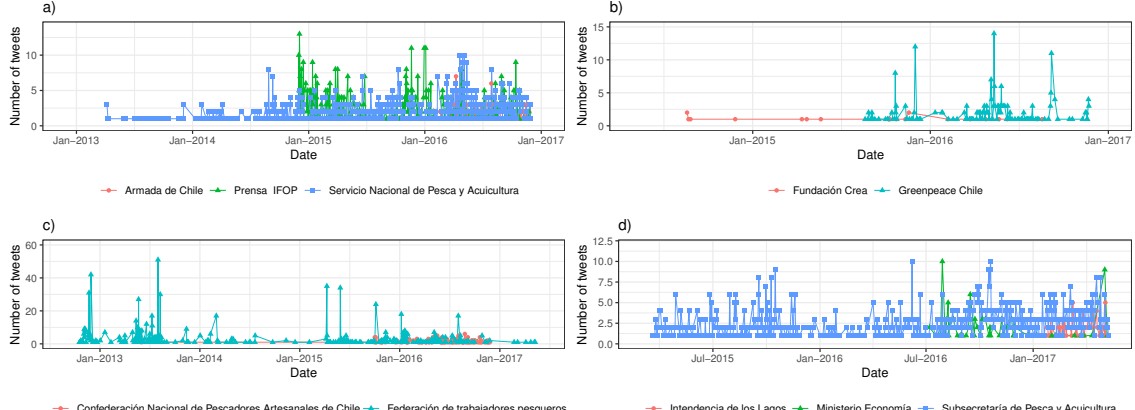

**Figure 4.** Time series of actors' utterances 2013–2017. (**a**) Number of tweets from Actors 1, (**b**) Number of tweets from Actors 2, (**c**) Number of tweets from Actors 3, and (**d**) Number of tweets from Actors 4.

As seen, several actors –particularly sectorial ones and Greenpeace– were already active in the analyzed topics before the beginning of the red tide crisis. For most of them, a spike can be detected as the crisis begins, although the general distribution does not differ significantly from the observed behavior before or after. There are, however, actors that start tweeting on the relevant topics of the corpus (see Table 3) as an effect of the crisis. Some of them are sectorial actors, such as the National Federation of Chilean Fishermen (Conapach) and the Fisheries Promotion Institute (IFOP), while others are not directly involved with the fishing sector, such as the Ministry of Economy (Minecon), local authorities (Intendencia), the Navy (Armada de Chile) and Crea Foundation. Noticeably, most of them remain active at tweeting relevant topics of the corpus months after the events, showing that Twitter users continue posting on the crisis long after the phenomenon passed, as argued by Ref. [43].

## 4.2. Actor-Centered Meaningful Components before, during and after the Crisis

In this subsection, we aim at identifying the basic meaningful components that characterize actors' experiences on different periods of the crisis. For this, we use the analysis of word clouds combined with egocentric networks of selected concepts. Figures 5–8 shows word clouds with the main keywords containing the utterances of four actors (technical agencies, NGOs, artisanal fishermen and political authorities) before (Figure 5), during (Figure 6), after the crisis (Figure 7), and in the whole considered period (Figure 8).

As shown, there are divergences and congruencies regarding the meaningful topics for each group. In Figure 5 (before the crisis), we can observe that "fishing" (pesca) and particularly "fishing regulations" (ley de pesca) are two of the most significant topics for technical agencies, fishermen and political authorities, while NGOs are not interested in the particular topics relevant for the island, but on general issues such as climate change and the melting of glaciers. Fishing regulations is a significant topic for artisanal fishermen and for the fishing industry as well because it defined several aspects regarding the political economy of the island, such as fishing sectors, fishing quotes, ban on fishing particular species and concession periods for the industry [131].

Interestingly, as shown in Figure 6 (during the crisis), the topics red tide (marea roja) or Chiloé disaster (desastre) become central for technical agencies and NGOs, while the topics fishing and fishermen are significant for artisanal fishermen and for political authorities. Even fishermen keep more or less the same of topics before, during and after the crisis, focusing on a critical discussion on fishing regulations affecting artisanal fishing and shellfish extraction. Certainly, the topic red tide turns out to be relevant for fishermen and political authorities as well, yet the former read the crisis as a problem of employment and subsistence (fishing), while the latter observes this as a political problem in need of an agreement (acuerdo). As Ref. [42] find out, Chiloé inhabitants and particularly fishermen are used to red tides. They know how to identify secure sectors for shellfish extraction and fishing. However, the dimensions and intensity of the 2016 red tide crisis were so extraordinary that there was no place to move the work activity. Thus, the red tide became a factual work prohibition.

As seen in Figure 7, while technical agencies, fishermen and political authorities return to topics related to fishing and fishing regulations as in the period before the crisis, the words "salmons", "Chiloé" and "government" remain as the most significant topics in NGOs discourse. Put in other words, NGOs enter late into the crisis discourse; they, however, preserve the key message of the crisis in the long run, thereby accomplishing a denunciative memory function that counts as a source for collective learning processes as to how to deal with this type of critical situations, as argued by Ref. [54,126]. These can also be seen by NGOs in Figure 8, where the words "Chiloé", "disaster" or "salmons" are clearly distinguishable as significant communication topics, while the other actors share fish-related words as central topics.

Word clouds are one of the easiest ways of visualizing both the meaningful boundaries of different discourses as well as the relevant topics the actors are interested in. A more detailed view on the meaningful construction of discourses is offered by egocentric networks. In Figures 9–11, we observe different networks of meaning construed from the perspective of relevant concepts of the crisis, such as artisanal fishing, red tide, dumping of salmons, and Sernapesca (the National Fishing Service).

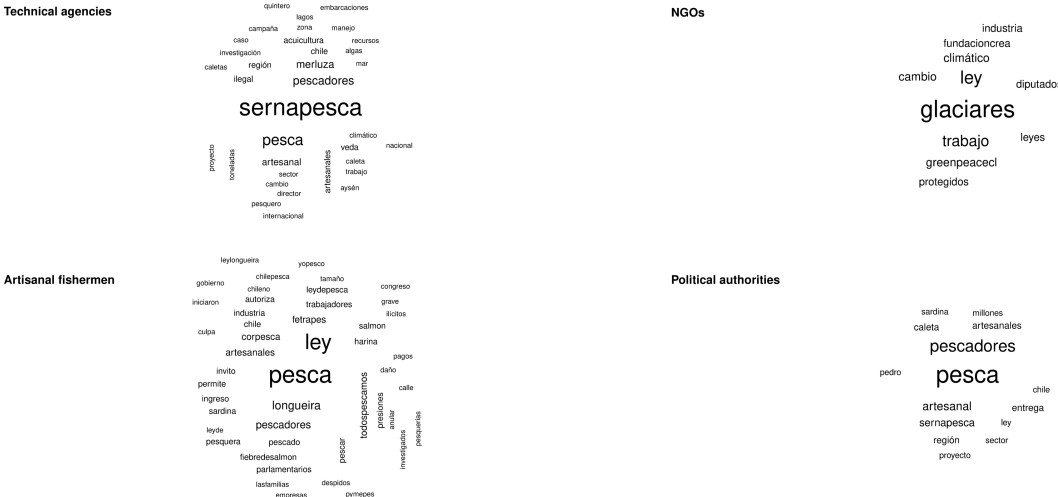

**Figure 5.** Keywords by groups of actors, 2013 to February 2016.

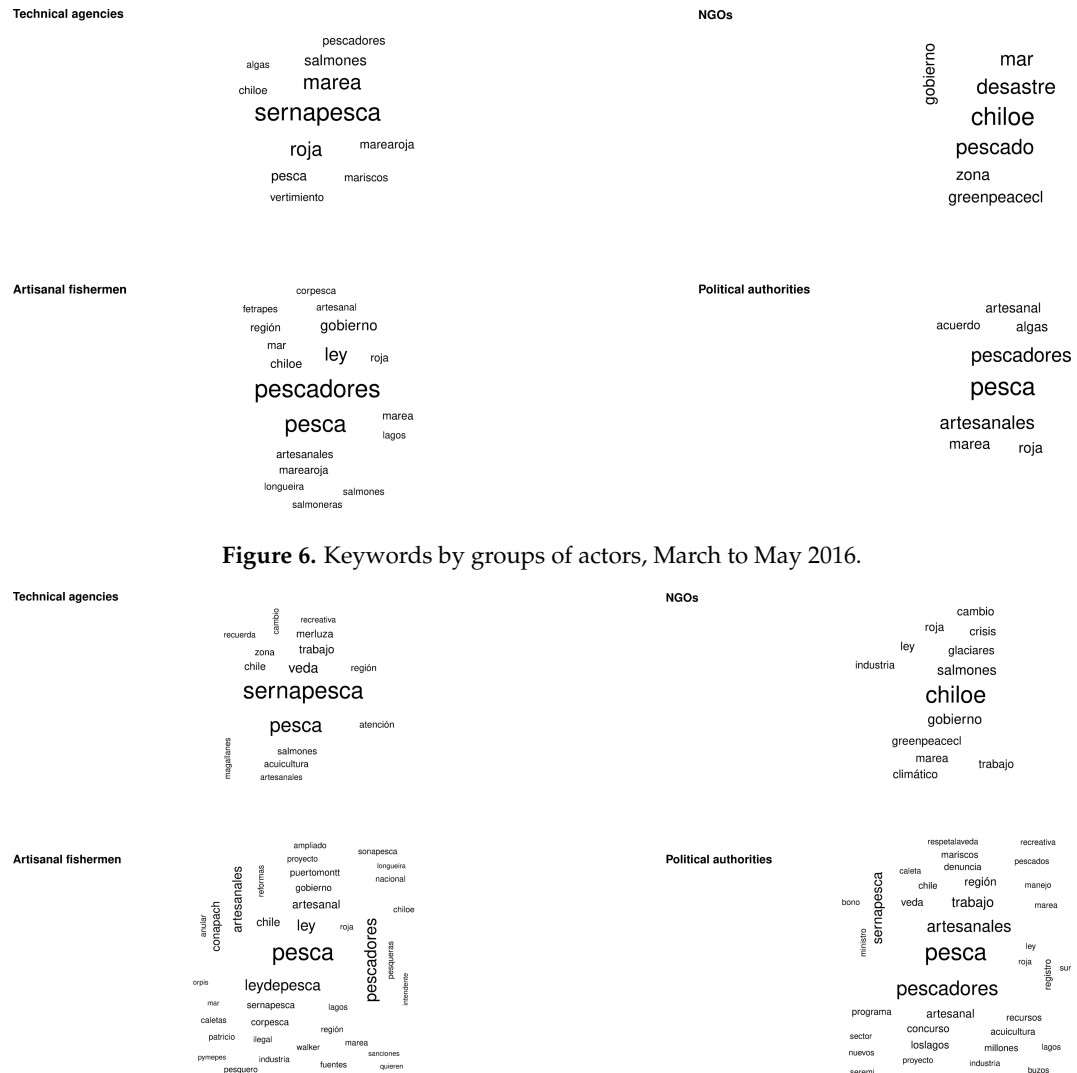

**Figure 6.** Keywords by groups of actors, March to May 2016.

**Figure 7.** Keywords by groups of actors, June 2016 to January 2017.

In Figure 9, before the crisis, we analyze two central concepts: artisanal fishing (pesca artesanal) and Sernapesca. Related to the concept of artisanal fishing (and fishing), we observe most of the topics which are relevant in the word clouds in the same period, yet we identify here the weight of relationships among terms. This allows us to build the limits of the general semantic structure of the period from a particular and localized social position. The focal points before the crisis are "fishing" and "Sernapesca". Fishing means "artisanal fishing" in the first place, as shown in Figure 9. It refers to a group of words concerned with sustainability problems experienced by artisanal fishermen, such as industrial fishing, illegal fishing, trawl fishing, dismissals from the industry, aquaculture, recreational fishing, and the fisheries law (leyde-pesca). When we compare this information with the word clouds, we notice that this web of meaning is mainly related to artisanal fishermen and to political authorities, while the network around Sernapesca in Figure 9 (the National Fishing Service) represents the communication of technical agencies before the crisis, which deals mainly with the bureaucratic functioning of the office and the role of seizing (incauta) illegal fishing.

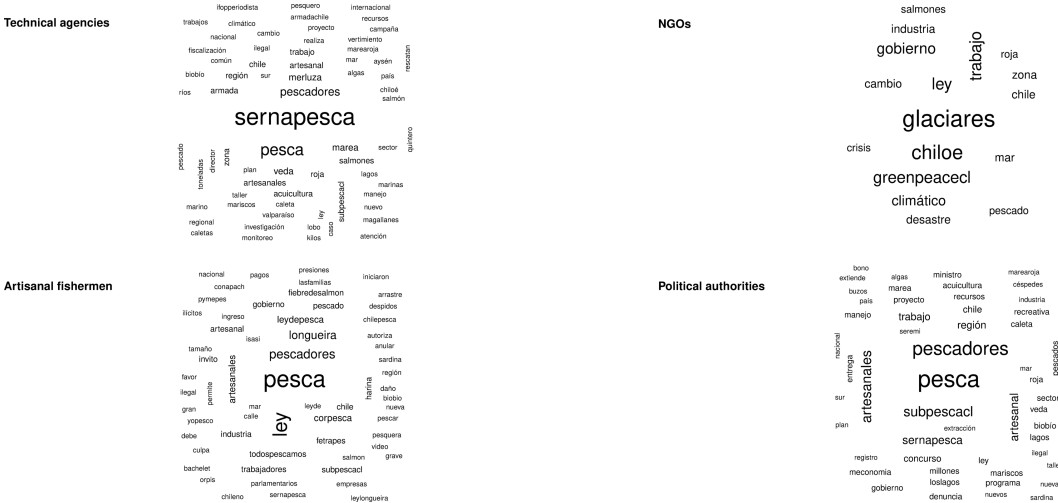

**Figure 8.** Keywords by groups of actors, 2013 to 2017.

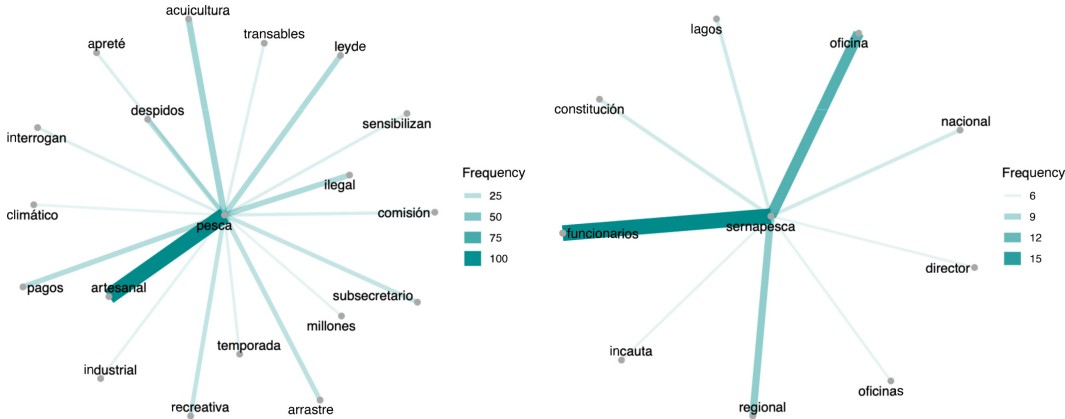

**Figure 9.** Egocentric networks by topics, 2013 to February 2016.

In the networks during the crisis (Figure 10), the concept of artisanal fishing remains as a significant vertex. However, its network changes and new relevant links are drawn to the nodes "commission" and "leaders", which represent fishermen in a series of negotiations with the government until the end of the crisis in May 2016. The other two networks are at the semantic core of the crisis: the vertex "red-tide" and "salmons", which reconstruct two crucial events in the development of the crisis. The former relates directly with the massive stranding of shellfish (vertex machas) on 25 April 2016, as well as with the "presence of the red tide phenomenon" conceptualized in the network as a "crisis"; the latter refers to the "dumping of tons of dead salmons" into the sea that took place 75 nautical miles off the northwest coast of Chiloé Island from 10 to 26 March 2016.

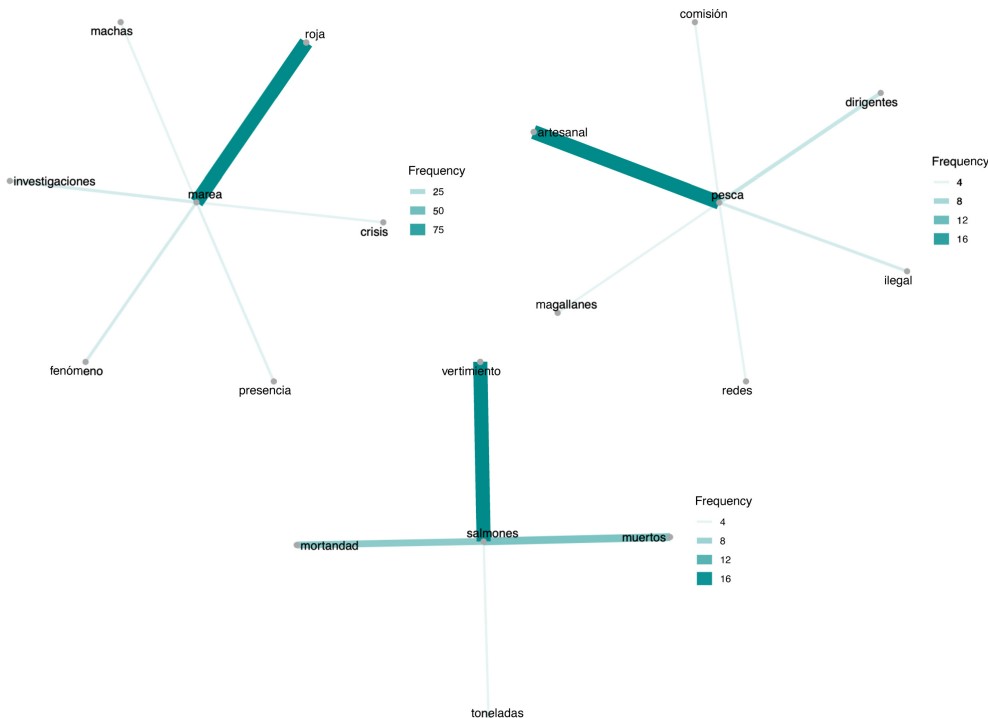

**Figure 10.** Egocentric networks by topics, March to May 2016.

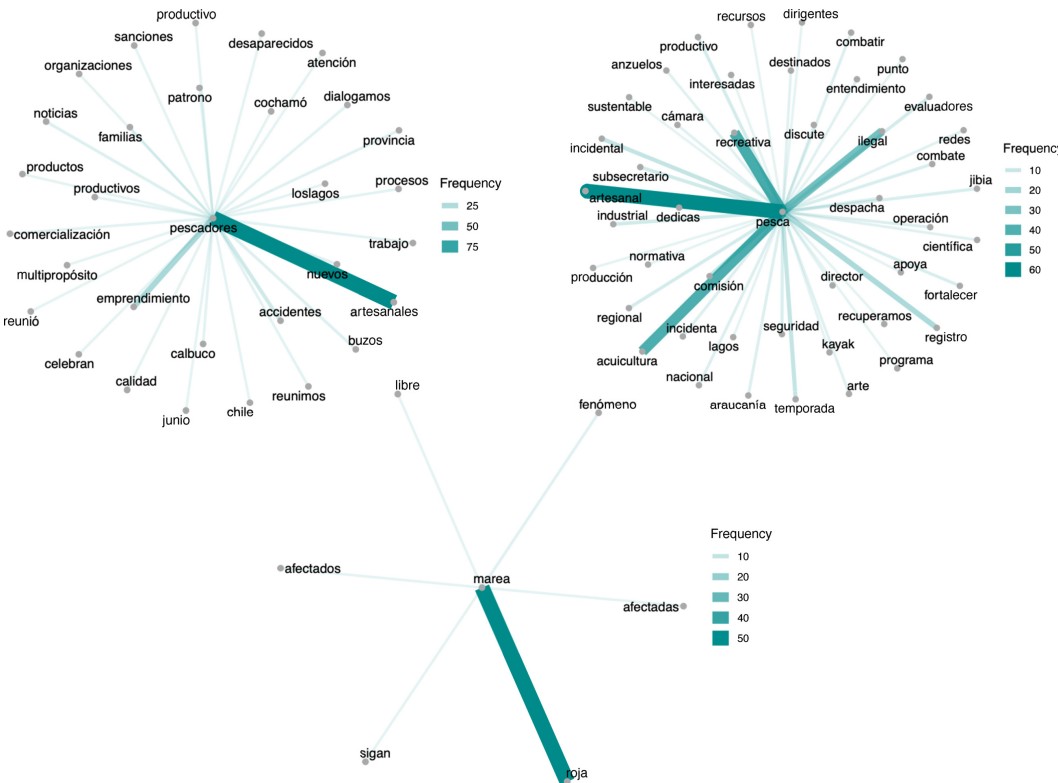

**Figure 11.** Egocentric networks by topics, June 2016 to January 2017.

In the networks after the crisis (Figure 11), we observe a focus on persons. While during the crisis the vertex "red-tide" refers to the phenomenon itself, after the crisis it deals with the consequences: the "affected" persons and areas, and the zones "free" from the algae. The focus on persons is confirmed by the vertexes "fishing" and "fishermen", which show how important is the artisanal dimension in social relations on Chiloé Island. While the vertex "artisanal fishing" draws its main links to problematic topics already present before the crisis (such as illegal fishing, aquaculture), the vertex "artisanal fishermen" focuses on the ways out of the crisis, either politically or economically. Politically, the vertex refers to the negotiations with the government after the crisis, including terms such as dialog, process, organization, attention, reunion, which denote some processes of collective learning. Economically, the vertex is closely related with the concept of entrepreneurship (emprendimiento). Belonging to this semantic family are also terms such as products, productive, commercialization and multipurpose, which mostly refer to the different economic activities fishermen carry out to cope with the crisis.

As seen, artisanal fishermen and artisanal fishing are central vertexes of meaning-making on Chiloé Island. The world is construed around them as they become a source of meaning for reconstructing social life (politically and economically) when a disrupting event such as the red tide crisis takes place. However, when we compare the webs of meaning for different actors during and after the crisis, we notice that only for fishermen artisanal fishing is relevant, as shown in Figures 12 and 13.

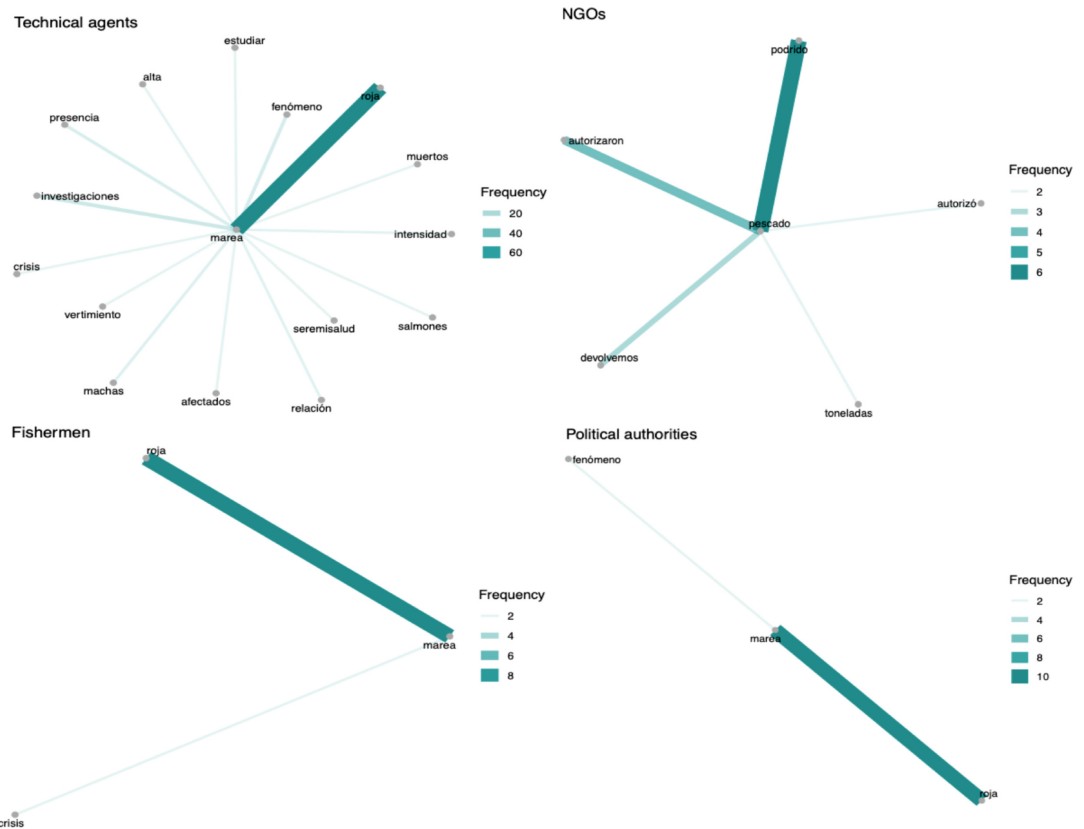

**Figure 12.** Egocentric networks by actors, March to May 2016.

In Figure 12, during the crisis, we observe that for every actor the vertex "red tide" is relevant. Nonetheless, while for fishermen the event is understood as a "crisis", political authorities consider it as a "phenomenon". This semantic difference reflects the ontological gap between the lived experience of the crisis and observing the crisis from a distance: it is a crisis when we experience it, and it is a phenomenon

when we observe it. A similar sense share NGOs: they are rather interested on the causes of the red tide as they reflect on the authorization of dumping rotten fish (vertex pescado podrido) into the sea on March 2016. They do not "live" the experience; they analyze it, but their work becomes crucial after the crisis for maintaining the memory of the facts and trigger learning processes as well as inform on support activities for fishermen. After the crisis (Figure 13), technical agencies and NGOs come along these lines, while only for fishermen is "artisanal fishing" the most relevant vertex organizing other meanings. When compared with Figure 11, we can conclude that it is their narrative, namely fishermen's narrative, what governs island's self-identity. Certainly, there might be conflicts in the construction of this identity as Ref. [45] have argued, yet if there is some identity on the island, fishermen have a significant role to play in it.

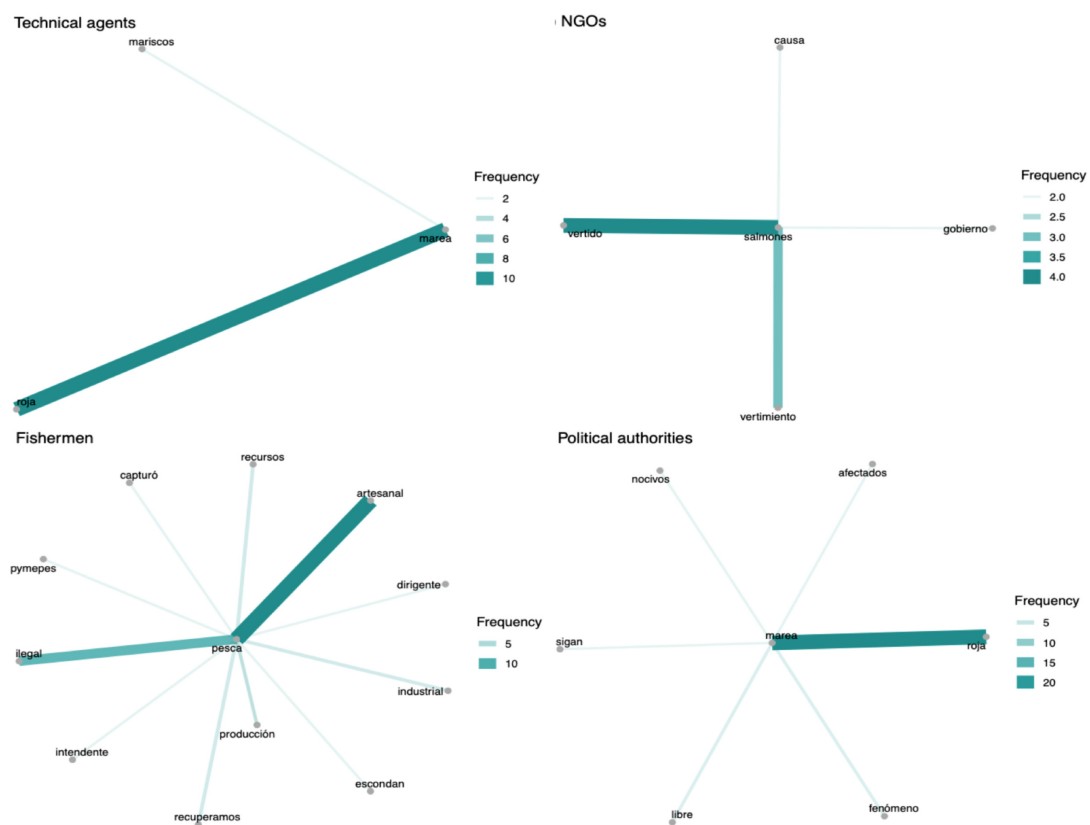

**Figure 13.** Egocentric networks by actors, June 2016 to January 2017.

*4.3. Constructing the General Meaningful Depiction: Semantic Clustering*

Having analyzed in detail the meaningful construction of different actors on the crisis through word clouds and egocentric networks, we aim now to increase the level of abstraction to observe the whole semantic structure of the periods before, during and after the crisis. We begin this analysis by using hierarchical clustering on the topics of our corpus (Figure 14).

In Figure 14a, before the crisis, we observe three themes that stand out from the rest. The first one (orange) refers to illegal fishing of sardine and the pressures for authorizing it. Contrasting this set with the word clouds, we can identify that mostly artisanal fishermen support this statement. The second group (light green) refers to the development of the salmon industry in the region, which has historically been object of a highly contentious debate about its environmental and social sustainability. The keyword "salmon fever" (fiebredelsalmon) refers to a documentary issued at that time precisely denouncing this

situation. Interestingly, the salmon industry will be on the spotlight in the period during the crisis because of the dumping of dead salmons into the sea [46]. This reveals that the industry is a permanent source of conflicts in the region.

The third group (blue green) concentrate terms related to the "fisheries law" or "Longueira law", after the minister promoting it, which was passed in February 2013 and originated political and social controversies as it was blamed for favoring industrial fishing and affecting artisanal fishermen. Tweets on the topic also alleged bribes received by congressmen from the fishing industry to approve the legislation.

Other topics are also present in this period (violet group). They turn later into key elements of the red tide crisis and the uprising of Chiloé inhabitants during May 2016. Tweets mention, e.g., artisanal fishing, artisanal fishing associations, unemployment; some of them mention possible threats affecting the island, including climate change, the ban of fishing for some species, sectorial projects to contain sea weed in the region, and even the practice of dumping dead fish into the sea, which positions as the most relevant topic in the next period.

In Figure 14b, during the crisis, there are two outstanding, yet expected topics: one refers to the relationship between the dumping of dead salmon into the sea and the red tide bloom; the second one concerns with the effect of algal blooms on fishing coves and aquaculture projects. Other words refer to different aspects of the crisis, such as the death of tons of fish, the economic "disaster" it produced on the fishing industry (and the salmon industry in particular), the surveillance work and prevention brought forward by the authorities to avoid intoxications, the stranding of fish and shellfish at Chiloé coasts on 25 April 2016 —a strong image which captured media attention for weeks—, and the different initiatives attempted by both the central Government and local authorities to contain the crisis and negotiate with local fishermen organizations. A mention is also made to the fisheries law, one of the most relevant topics of the previous period. As we can observe, the red tide crisis dominates completely the semantic scenario, thereby transforming communication into a rather unidimensional realm.

In Figure 14c, after the crisis, we can observe an increase in the diversity of themes, which seems to be signaling that the communication is going back to some type of meta-stability. While the singularity or one-dimensionality of communication is over, there are seems to be a general meaning pointing out to the reconstruction after the crisis. To this meaning contribute the focus on "recovery" (training programs and programs supporting local entrepreneurship), other mentions aimed at improving monitoring, surveillance, management of fishing resources and the training of local fishermen—all of them pointing out to collective and organizational learning processes. One of the key themes is Sernapesca's campaign to observe fishing prohibitions. Some other topics call for a larger change and to advance new legislations.

Maybe the most interesting result here is the new role taken up by the expression "red tide", which raised to a highly condensed semantic linking up to the nexus of ideas and themes that made up the crisis itself. The concept acts as a kind of "semantic memory" or trace of the crisis, yet apparently unconnected from other themes. The previous findings are confirmed when looking at how the tweets cluster across the three periods. Before the crisis, three thematic groups of tweets can be distinguished. The crisis produces a reduction in the variety of themes, which cluster more tightly together. While after the crisis, the themes multiply.

In Figure 15a, before the crisis, we observe three groups of tweets. Cluster 1 (red) and 2 (green) present a similar participative structure of the four actors, with a less dense participation of NGOs and an intense participation of fishermen. We also find a relevant participation of political authorities in cluster 3 (blue). Since this technique clusters tweets (and not topics), in each case we have analyzed the contents of nine nearest tweets to the cluster's centroid (see Appendix A), in order to identify the thematic profile of clusters: cluster 1 refers mainly to regulations and risks, cluster 2 refers to surveillance activities, and cluster 3 informs on activities of political authorities in relation to fishing topics.

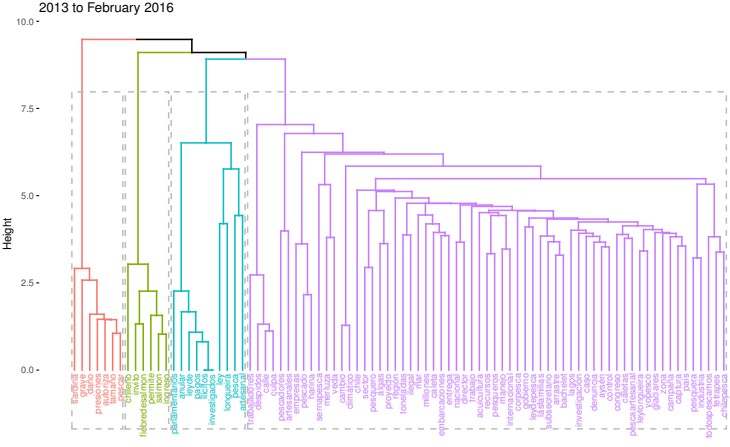

(**a**) Before the crisis: 2013 to February 2016

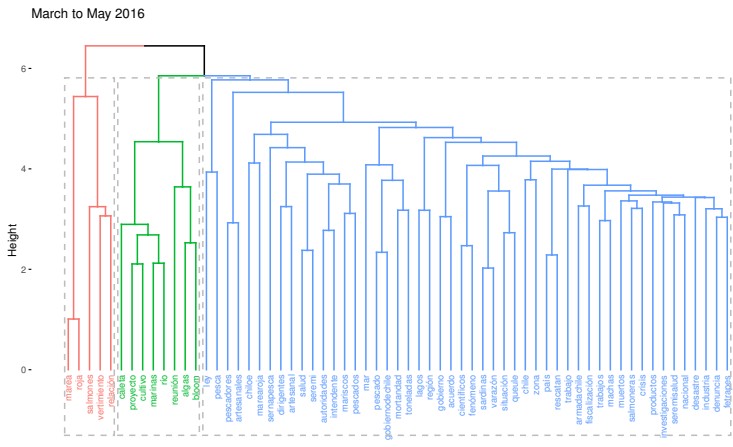

(**b**) During the crisis: March to May 2016

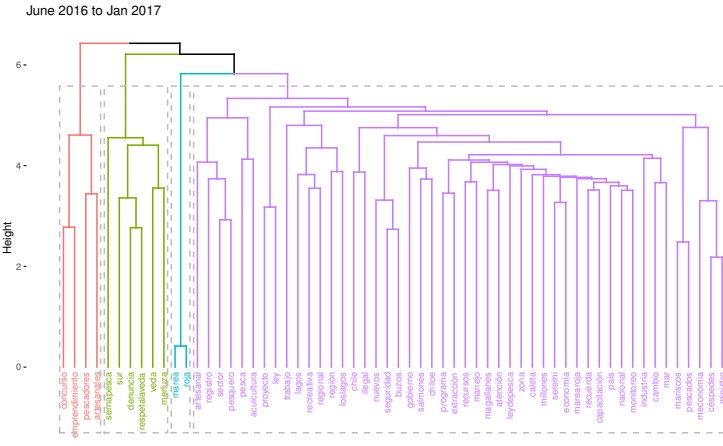

(**c**) After the crisis: June 2016 to Jan 2017

**Figure 14.** Hierarchical clustering of topics (**a**) 2013 to February 2016; (**b**) March to May 2016; and (**c**) June 2016 to January 2017.

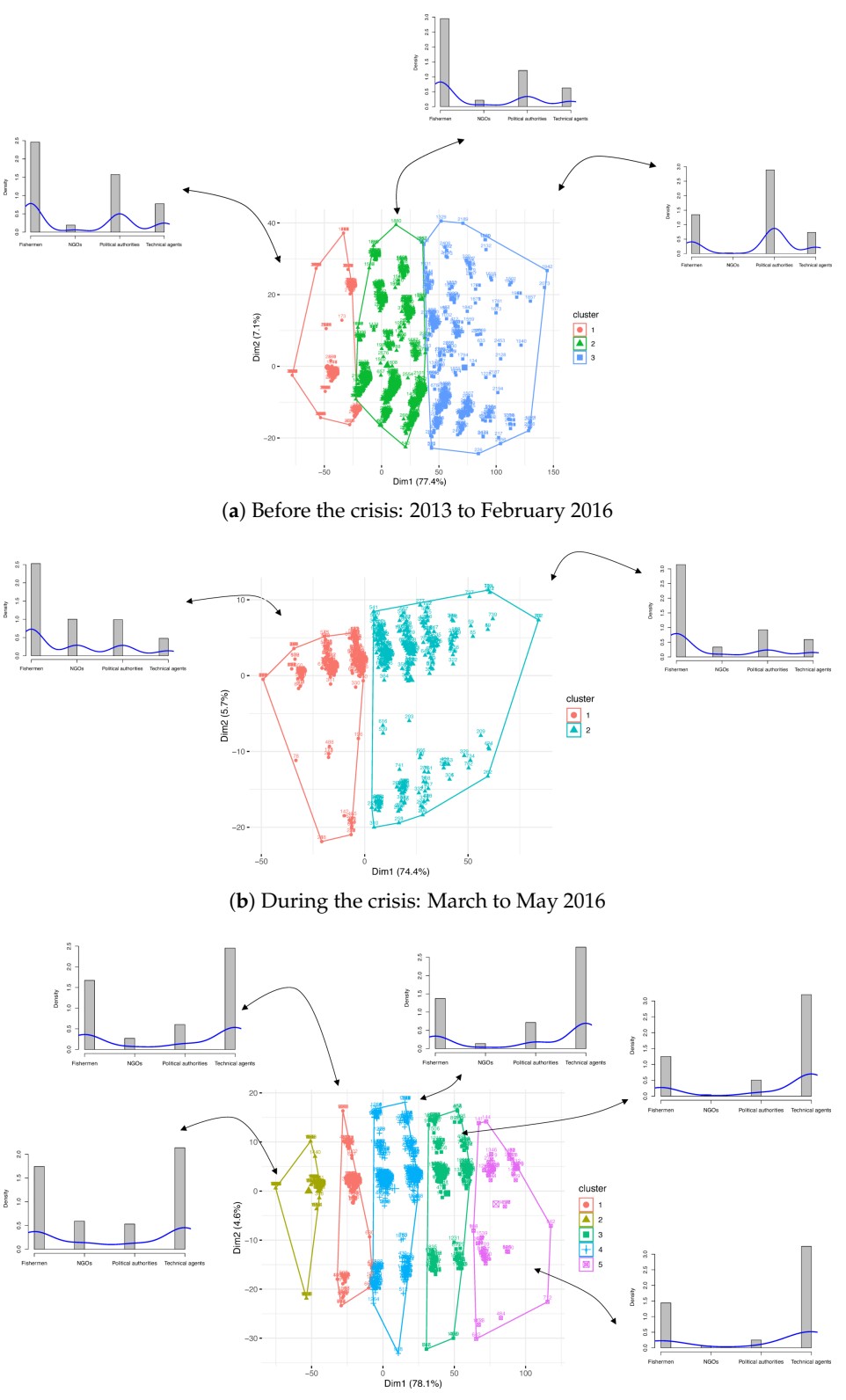

(**a**) Before the crisis: 2013 to February 2016

(**b**) During the crisis: March to May 2016

(**c**) After the crisis: June 2016 to Jan 2017

**Figure 15.** Cluster representation with k-means (**a**) 2013 to February 2016; (**b**) March to May 2016; and (**c**) June 2016 to January 2017.

Figure 15b, during the crisis, reduces communication to two clusters: cluster 1 (red) is smaller but presents a more normally distributed participation of actors, while in cluster 2 (green) participation is rather irregular. Both clusters are dominated by the discourse of fishermen, while NGOs' discourse is denser in cluster 1 than in 2. When analyzing tweets (see Appendix A), cluster 1 refers mostly to negatively-loaded information on the red tide, while cluster 2 refers mainly to information on surveillance activities.

After the crisis (Figure 15c), communication is distributed into five clusters. Interestingly, the discourse of technical agencies prevails in the five clusters, counterbalanced in some cases by the discourse of fishermen. In terms of density, NGOs are relevant only in clusters 1 and 2 and rather insignificant in the other clusters, while political authorities present a similar density in four clusters, with the exception of cluster 5. The main references of each cluster are the following. Cluster 1: study on salmons; cluster 2: training programs; cluster 3: information on official activities; cluster 4: fishing management; cluster 5: political support for fishermen. As seen, after the crisis there is a more diverse communication, yet all of them are focused on different recovery actions and information.

### 4.4. Topological Distribution of Crisis Communication

While in Figure 15, we become aware of the density of participation of each actor within a cluster, in Figure 16 we can observe the topological distribution of communication for the different actors, before, during, after the crisis, as well as in a general picture of the whole period. With this topological distribution based on a principal component analysis, we have a view on the semantic distance and proximity actors' utterances have with each other and how they change in time (Figure 16).

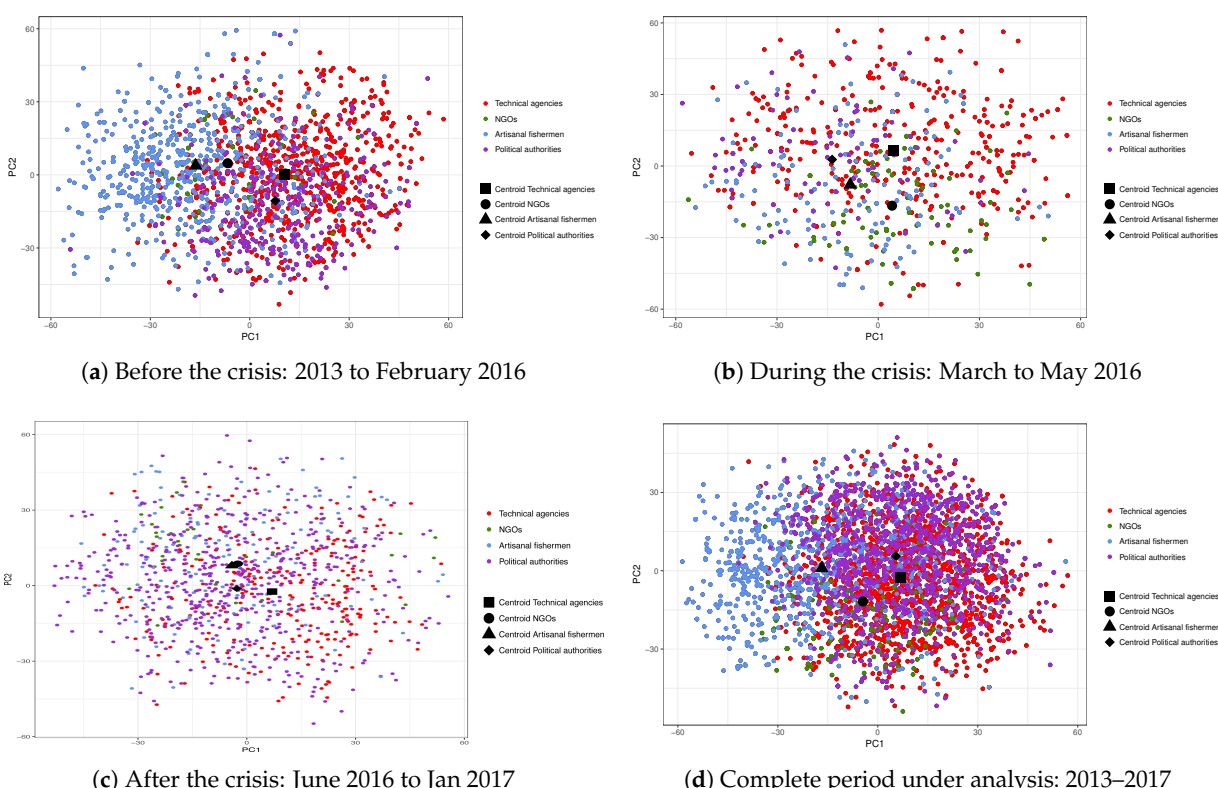

(**a**) Before the crisis: 2013 to February 2016

(**b**) During the crisis: March to May 2016

(**c**) After the crisis: June 2016 to Jan 2017

(**d**) Complete period under analysis: 2013–2017

**Figure 16.** Topological distribution of utterances by groups of actors: (**a**) 2013 to February 2016; (**b**) March to May 2016; (**c**) June 2016 to January 2017; and (**d**) 2013 to 2017.

Comparing the scope of communication brought forward by the four groups (Figure 16), we may notice that before the crisis (Figure 16a) an important group of fishermen's utterances are located in a particular topological area where they do not intersect with communications of the other actors. Centroids also reveal this particular positioning. While artisanal fishermen are far left in the topological area and near to NGOs' communications, technical agencies and political authorities share similar positions right of the topological space.

During the crisis (Figure 16b), communications are rather dispersed in the topological space. However, when observing centroids, the opposition that takes place between technical agencies and NGOs becomes interesting: the former dominates the superior ranges of the space, while the latter are mostly located in inferior ranges. As we know, both actors confronted publicly regarding the responsibility of the salmon industry and, particularly, the dumping of dead salmons into the sea [42,47]. While technical agencies attributed the red tide mainly to natural causes, NGOs put emphasis on the accelerating role played by the dumping of dead fish into the sea on March 2016. This approach has been supported by different viewpoints, namely, court decisions and scientific analysis (see [46,132]).

After the crisis (Figure 16c), a significant coincidence between the communications of fishermen and NGOs seems to emerge (visible in the two overlapping centroids). From Figure 15 we notice that this coincidence refers to clusters 1 and 2, in which the study of salmons and training programs for fishermen are the central topics. This is interesting because when we pay attention to Figure 16d (the topology of the three periods), fishermen and NGOs present fairly separate centroids, which means that they originally have quite different interest but become closer with the crisis. Additionally, we observe that fishermen and technical agencies, though overlapping at some points, have also diverging interests that we recognize in the opposite position of their centroids as well as in the general location in the topological space: left-leaning position of fishermen, right-leaning position of technical agencies. To that extent, while some utterances may intersect at some points and topics between fishermen and technical agencies or between fishermen and NGOs, the different positioning of these actors in the topological space reveal rather dissimilar experiences and diverse comprehensions on the crisis. As argued by [45], here we can find an explanation for the difficult self-recognition of the island.

## 5. Discussion

In this article, we have argued that the analysis of social media in social-ecological crises and, particularly, Twitter analysis, when performed through a multimethod approach and interdisciplinarily examined, offers (a) a structural systemic view of the semantic boundaries and meaningful networks of a social-ecological crisis, (b) an involvement with actors' meaning-making, and (c) an overview on the learning processes actors and organizations undergo in a social-ecological crisis situation. In order to address these points, we organize our discussion into three main subsections: sources of meaning-making, dynamics of meaning-making, and breakdown of meaning and learning processes in crisis situations.

### 5.1. Sources of Meaning-Making

One primary and useful performance of word clouds applied to different social actors is to clearly define the central and peripheral concerns of their discourses as well as the distance and overlaps that these concerns have among those actors. When, additionally, we have a long time series, we can identify how discourses and concerns change in time and how actors become complexly intertwined along the process. From previous research [42,44,47,126], we know that different actors construe different narratives on factual events. Word clouds show this, yet they also show that the general concept of fishing is a keyword for every actor with the exception of NGOs (Figures 5–8). Fishing is a meaningful source for actors concerned with the island.

From the interpretative viewpoint of word clouds combined with egocentric networks, the semantic family grows: terms such as fishing, artisanal fishing, artisanal fishermen become clearly sources of meaning-making for Chiloé inhabitants as well as for some actors concerned with sustainability problems of the island (industrial fishing, illegal fishing, trawl fishing). That these terms are sources of meaning implies that the meaningful experience of actors is organized around those terms. Actors' experience could be different, as shown in word clouds and egocentric networks, but the focal point of fishing organizes the experience for almost every actor. This is particularly true for fishermen. For them, daily life and extraordinary events such as the 2016 red tide crisis are observed through the general prism of "fishing", so that in regular times fishermen are concerned with regular problems (as said, illegal, industrial fishing) and in times of crisis they become concerned with the effects of the crisis for artisanal fishing, i.e. with the effects of the crisis for their employment.

Supplementarily, we can observe in word clouds and networks that alliances between actors are not permanent in time. They change according to the circumstances. Fishermen and NGOs are not part of the same meaningful world before the crisis, yet they become allied during and after it. Political authorities and artisanal fishermen can agree that "fishing" is the most relevant topic on the island yet on different grounds, either because of livelihood or political instability. Meaning is, therefore, the basis for constructing political alliances during a social-ecological critical event as the 2016 red tide crisis.

With clustering techniques, we become aware of the general semantic structure of the analyzed period. We are allowed to observe how concepts uttered by actors construct meaningful communication topics that differentiate from each other in semantic patterns before, during and after the crisis. Interestingly, with a focus on the period before the crisis, the major cluster with loosely coupled elements (Figure 14a) already contains topics that become relevant in the periods during and after the crisis. To that extent, it could be useful to deeper analyze loosely coupled semantic components in cluster representations in order to anticipate possible actors or topics involved into a crisis that might eventually develop later on.

While word clouds and egocentric networks are crucial to correlate actors with discourses in terms of distance and overlaps, one of the major advantages of clustering techniques is that we can easily identify the internal and external boundaries of communication processes. Internal boundaries are defined by the aggregation of single words pointing out to a meaningful behavioral pattern such as the dumping of dead salmons into the sea on March 2016 (Figure 14b) or the focus on "recovery" after the crisis in Figure 14c (training programs and programs supporting local entrepreneurship), while external boundaries of communication are identified in the diversity of topics into the loosely coupled section of clusters. In this vein, clustering connects the microworld of particular meanings of each word with the macroworld of discourses that arise when words semantically interact with each other in time.

Supplementing this actor-centered and also structural construction of meaning, principal component analysis applied to Twitter communication offers an additional perspective to the analysis of social ecological-crises. It gives us a topological representation of actors' communication along the analyzed periods. A crucial finding in this regard is the fact that a significant number of fishermen's communications are reported in a position that does not overlap with the utterances of any other studied actor, neither with technical agencies, NGOs, nor with political authorities. This means that a group of fishermen's expectations are, to some extent, radically different from other actors' expectations, while another group of fishermen present overlapping communications with other actors.

Two problems for island's cohesive social life arise from this. Firstly, if this is so, even within the artisanal fishermen there are relevant differences in expectations, so the construction of a shared identity on the island is not an easy task; and secondly, there could be an epistemological barrier between the fishermen located on one side of the topological space and the rest of the social world, including fishermen whose communications overlap with those of other actors. In that sense, there is an unstable social basis for collective action on the island.

## 5.2. Dynamics of Meaning-Making

As described in our literature review, there are two main fields in which Twitter activity in crisis situations has proven useful: first, the identification of competing interpretations on the crisis, and second, the dynamics of crisis communication evolving from a critical threshold in which uncertainty prevails to a more reflexive stage where the critical event is retrospectively re-construed and lessons are learned. The question rises whether these two main insights of Twitter communication research are recognizable in our findings. In this subsection, we discuss the first point.

In a press analysis, Ref. [47] have identified two narratives on the red tide crisis: the narrative of fishermen and Chiloé citizens who emphasize the responsibility of the salmon industry in the crisis, and the narrative of media, which observe the crisis as a natural phenomenon and as a problem of public security. Based on in-depth interviews and press analysis, Ref. [42] identify four narratives: technical agencies, NGOs, fishermen and political authorities. While NGOs' narrative (human made contribution to the red tide) discuss technically with technical agencies (natural origin of the crisis) and make alliances with fishermen, political authorities support the discourse of technical agencies and are concerned with public security on the island. Based on in-depth interviews and press analysis as well, Ref. [44] identify two narratives: the demand for redistribution within the modernization project, and the call for a redefinition of the concept of development on the basis of the identitarian uniqueness of the island. In our research, we follow the classification of actors identified in Ref. [42], however, we can now register different overlaps among actors' discourses which allows for a more dynamic consideration of meaning-making.

As we can observe from word clouds and egocentric networks, the semantic family of "fishing" is a general source of meaning for most of the actors with the exception of NGOs. This does not mean, however, that there are two narratives; it means rather that from this original point different views are construed that change in time. Before the crisis, conversations on fishing are related to problems with the industry and to illegal fishing, while during and after the crisis the problem of fishing is the unemployment of artisanal fishermen. At least in our data, we do not observe that for political authorities the main concern had been public security as media inform: they are rather concerned with fishing and fishermen as much as fishermen. Certainly, both actors experience the crisis differently: fishermen experience it as a "crisis" and political authorities as a "phenomenon" (see Figure 12), yet both of them agree that they are dealing with an extraordinary event and share the concern on the semantic family of "fishing" (vertex "artisanal fishing" in fishermen, and vertex "affected-red-tide" in political authorities in Figure 13).

The dynamics of NGOs' discourse is highly interesting. Before the crisis, they are concerned with topics outside the semantic boundaries of fishing (glaciers, climate change and alike). As the crisis begins, however, they enter rapidly into a controversy on the origins of the red tide with technical agencies. While technical agencies (supported by political authorities) claimed that the red only had a natural origin (as argued by Ref. [41]), NGOs contended that the dumping of dead salmons into the sea on March 2016 played a relevant role in fueling the crisis (as recently argued by Ref. [46]). To that extent, NGOs became allied with fishermen during the crisis, yet not on the grounds of a concern with the general problem of fishing, but because of the development of the natural disaster. After the crisis, NGOs remain informing on the responsibility of the salmon industry in the crisis. They play a relevant role as a collective memory of the facts. Interestingly, this case confirms that actors do not need to share a basic semantic structure to become political allies in crisis periods; they rather require converging interests that account for a positive sum game.

The case of technical agencies is similar. They are concerned with "fishing" during the three periods; this is their "official" concern. Yet, while before the crisis they share several topics with fishermen (e.g., ban on fishing, artisanal fishing), during the crisis they enter into the controversy with NGOs on the origins of the red tide and, in this vein, their position becomes opposed to that of fishermen. However,

after the crisis, technical agencies are the most active actor in spreading information on recovery programs for fishermen, as observed in Figures 14c and 15c. In a different way to NGOs, technical agencies do not insist on uncovering the origins of the crisis; they rather move to support fishermen in the aftermath of the crisis.

Put in other words, the dynamics of semantic meaning-making is not predefined once for all in social-ecological contexts; it changes according to emerging events in the development of the crisis. Considering our literature review, our findings confirm that there are competing interpretations of the crisis, yet we demonstrate that these interpretations are not substantially tied up to actors as if they were elemental truths not affected by historical events. Actors' interpretations change according to circumstances; they either approach or move away from each other depending certainly on commitments but also on the sequence of events and political convenience.

*5.3. Breakdown of Meaning and Learning Processes*

Regarding the second general point in our literature review, namely, the statement that crisis communication in Twitter activity evolves from a breakdown of meaning at the core of the disaster to a more reflexive understanding of what happened, our findings reveal some interesting insights.

We agree with the literature that in the first moments of the crisis, the event itself –in this case, the red tide– dominates the semantic scene. Our cluster analysis of Figure 14b proves this clearly as the two outstanding topics refer to the red tide: the first one to the causes (dumping dead salmons) and the second one to the manifestations (algal blooms in aquaculture), while other loosely coupled words are concerned with visible and future effects. However, there is no evidence in our findings that red tide outbreak produced a breakdown of meaning. Rather, the reflexive work of making sense of the crisis begins right after the major bloom affecting the interior and exterior coasts of the island [42]. The controversy between NGOs and technical agencies, which continues throughout the core of the crisis, is an indicator of the reflexive tone of communication. There is, thus, no breakdown of meaning, but rather an intensive work on elucidating the causes and possible effects of the critical situation.

On the other hand, as said earlier, the semantic family of "fishing" is a constant source of meaning before, during and after the crises. The focus on "fishing" as a semantic source does not stop as the crisis breaks through, it only shifts the semantic target: from the concern on illegal and industrial fishing it moves to a concern on the impossibility of fishing because of the red tide, and thus, it turns into a problem of unemployment for fishermen. In this vein, the fundamental source of meaning ("fishing") remains, and it helps to make sense of the disruptive scenario from the beginning of the crisis situation. To that extent, the communicative context does not become "more reflexive" after the critical moment; it remains reflexive, yet with another purpose: the work on recovering from the impacts of the crisis.

An advantage of Twitter data is that we can capture meanings from the past. In so doing, we are allowed to see continuities and discontinuities of meaning in time, as we have proven in this discussion. Yet, particularly with some techniques it is possible to observe how future problems incubate. Figure 14a, before the crisis, shows in one particular cluster that the salmon industry has been a regular source of problems for Chiloé island, and Figure 15a shows in cluster 1 that the red tide and natural disasters were a concern before the crisis. Of course, this does not anticipate the breakout of the crisis, but it reveals the continuities and discontinuities of real problems affecting Chiloé inhabitants as well as the consciousness of the risks those problems involve. Relevant social-ecological crises present long-lasting incubation periods; they do not emerge from nothing. Put succinctly, to observe relevant clusters of communication in the present could offer a view on future relevant crisis. The construction of early warning systems could profit from this fact.

Because of these continuities and discontinuities as well as the shifts in meaning making and political alliances that we have detected, the construction of a solid collective learning process seems to be rather difficult. This is also the stance of the literature on the case [44,45,126]. Nonetheless, some facts speak for the possibility of these processes. The memory function of NGOs as a reminder of the responsibility of the salmon industry in the red tide is a valuable resource for collective learning; the experience of fishermen regarding the negotiations with the government in May 2016 is another resource; and the loosely coupled networks that develop after the crisis between political authorities, technical agencies and artisanal fishermen concerning the recovery programs are a third valuable resource for collective learning processes.

While this is true, our data also present a troubling fact. In Figure 16d, we observe that a relevant part of fishermen's communications locates quite apart from the rest. We can also see that fishermen's centroid show a significant distance with the centroids of other actors' communications in the topological space. This not only means that an important group of fishermen does not share opinions and topics with other actors, but also that fishermen themselves are divided into a more radical and a more integrated group. This correlates with the findings of Ref. [44] that on the island coexist two narratives, an identitarian narrative claiming for autonomy and a modern narrative demanding redistribution; and also explains the difficulties for self-recognition highlighted by Ref. [45].

This divide is, probably, the most relevant barrier against the construction of a solid collective learning process. It is true that there is a general semantic basis for constructing collective meaning, i.e. the concept of "fishing" and its derivatives, which comes to the forefront when a major crisis breaks out and turns into the fundamental motivation for collective action. Yet, the important convergence of fishermen with NGOs during and after the crisis (Figure 16c) dissolves when we pay attention to the whole picture (Figure 16d). So, the problem is not solved with a voluntaristic integration of local actors into decision-making processes. In our view, the question remains whether the problem can be solved at all.

## 6. Conclusions

In this article, we have shown that a social media approach to social-ecological crises can offer an actor-centered meaningful perspective on social facts as well as a systemic view of the communication dynamics before, during and after the crisis. We also have shown that while sources of meaning-making may remain in time, the dynamics of meaning-making shifts according to actors' commitments, the sequence of events, and political conveniences. Additionally, we have discussed the statement in the literature that the outbreak of the crisis relates with a breakdown of meaning. At least in the analyzed case, we observe a permanent and intensive reflexive work on elucidating the causes and effects of the crisis. Several conclusions can be drawn from our analysis. We describe them in detail.

First, a fundamental resource for constructing political alliances, learning processes and reflexiveness in social-ecological crises is to share a common source of meaning-making for different actors. In our case study, this common source is the semantic constellation of 'fishing' with all its variances (artisanal fishing, industrial fishing, illegal fishing, trawl fishing, ban on fishing, fishing law, fishermen). This constellation connects different and even opposite views and interests, yet it turns into a common ground for discussing and acting within a shared semantic framework that facilitates collective action for and against. While there might be conflicts of goals and concerns, they are processed within the semantic constellation of fishing.

Second, that actors share a semantic constellation does not mean, however, that conflicting interests and worldviews cannot be expressed. Social diversity lies at the basis of social-ecological systems [17]. There is, for example, contradictory performative effects between fishing and ban on fishing, or between artisanal fishing and industrial fishing. These rather opposite positions lead to political alliances (such

as those between fishermen and NGOs criticizing the salmon industry), generalized conflicts (between technical agencies and the government, on the one side, and fishermen and NGOs, on the other), and partial consensus (between fishermen and the government) on the causes and development of the crisis as well as on the measures supporting affected fishermen. To that extent, we confirm previous findings in the literature regarding the existence of competing interpretations in crisis processes.

Third, the force of social diversity emerges not only from the different structural positions that the involved actors occupy, but also from the dissimilar historical experience actors have regarding prior critical situations. Our results show (see particularly Figure 16) that fishermen do not communicate as a homogeneous actor. They are divided into two groups: while one group shares discursive contents with other actors, the other one has a discourse of its own. Drawing on previous research, we attribute this divide to different political stances on Chiloé Island: the former demanding redistribution of costs and benefits of modernization, and the latter emphasizing the uniqueness of Chiloé's identity [44]. This is a rather regular dynamics in accelerated modernization processes in which local groups follow different paths. In these cases, the divide is an obstacle that has to be overcome by collective action and learning processes [20,114].

Fourth, besides the competing interpretations in crisis processes, the literature stresses the breakdown of accustomed structures of meaning at the initial stages of the crisis. In our findings, there is no evidence supporting this. Affected actors begin with the reflexive work of making sense of the crises right after the major algal bloom. They look for the causes, speculate on who is responsible for the disaster, publicly accuse involved actors, organize themselves and demand solutions. It is true that there is a regime shift [133,134] in the main topic of communication after the outbreak of the red tide crisis (from 'fishing law' to 'red tide' and 'fishermen'), but there is no breakdown in meaning-making. In fact, communication is supported by the semantic constellation of 'fishing', i.e., it continues informing on topics that are relevant for the fundamental source of meaning-making of island inhabitants. This quick response shows the reflexiveness of local actors, particularly fishermen and NGOs, who immediately engage in making sense of the new situation, materializing it into political actions and triggering learning processes.

Fifth, continuities and discontinuities in meaning-making are a relevant factor that learning processes have to consider. Collective learning from social-ecological crises does not mean that every actor has to agree with the one and only interpretation of the crisis. This would involve an important loss in social diversity. Rather, collective learning means increasing the capacity of self-organization for resilience and sustainable development particularly in local communities and vulnerable areas [17–19]. With the 2016 red tide crisis in Chiloé, the local community has learned (or has confirmed previous learning) that political alliances are useful for bringing local problems to a public forum, they have developed the experience of negotiating with the government, and they have assembled a memory of the crisis through locally based social organizations that works as a political and normative repository for future events. For example, in the current COVID-19 pandemic, Chiloé Island is one of the least affected territories in Chile in terms of the spread of the virus, with 44 out of 14,139 active cases nationwide as of 8 October 2020 [135]. In co-ordination with local authorities, the community has self-organized to control access points to the island and to inform the inhabitants on the preventive measures. The experience with recent crises seems to work in the current critical situation. Further research can address this point.

Finally, research on the 2016 red tide crisis on Chiloé Island, the most significant algal bloom in the southern hemisphere in the last century, has been mainly conducted through qualitative methods (in-depth interviews, framing methods). In this article, we applied a multi-method approach to Twitter data through which we have supplemented and discussed existing findings in the literature and produced new insights on the 2016 red tide crisis on Chiloé Island, Chile. Our article supports the argument, brought forward by multidisciplinary scholarship in resilience research, disaster risk reduction, neogeography and social-ecological systems, about the need of developing and field-testing mixed methods that connect semantic analysis with the new opportunities offered by social media, voluntary geographic

information and other data sources. Taking advantage of the new opportunities increasingly made available by current innovations in information technology is crucial for improving our understanding of social-ecological systems. Accomplishing this depends on the research design (different actors' views distributed in a long-time span) and on the combined character of the techniques applied. Good qualitative research can be thus supplemented with new possibilities of capturing meanings from the past that cannot be retrieved otherwise. This is particularly relevant for studying social-ecological crises and supporting collective learning processes that point out towards increased resilience capacities and more sustainable trajectories in affected communities.

**Author Contributions:** Conceptualization, A.M., P.A.H., M.B., G.A.R.; methodology, G.A.R., P.A.H.; software, G.A.R., P.A.H.; validation, G.A.R., P.A.H.; formal analysis, A.M., P.A.H., M.B., G.A.R.; investigation, A.M., P.A.H., M.B., G.A.R.; resources, A.M.; data curation, P.A.H.; writing–original draft preparation, A.M., P.A.H., M.B., G.A.R.; writing–review and editing, A.M.; visualization, P.A.H.; supervision, A.M.; project administration, A.M.; funding acquisition, A.M., G.A.R. All authors have read and agreed to the published version of the manuscript.

**Funding:** This research was funded by Fondo Nacional de Desarrollo Científico y Tecnológico, ANID FONDECYT grant number 1190265, and ANID FONDECYT grant number 1080706.

**Acknowledgments:** Authors thank to Claudia Alonso for Figure 1 and to three anonymous reviewers of *Sustainability* for helpful comments.

**Conflicts of Interest:** The authors declare no conflict of interest.

## Appendix A

**Table A1.** Contents of the nine nearest tweets to cluster's centroid: 2013 to February 2016.

| Cluster 1 | Cluster 2 | Cluster 3 |
|---|---|---|
| Sernapesca decomisó 400 kilos de salmón sacados del Río Petrohué | Sernapesca y Carabdechile incautan mil 400 kilos de centolla en Puerto Natales. #Magallanes. | IFOP se reúne con pescadores artesanales de Quintero para exponer proyecto tras derrame del 2014 |
| Sernapesca informa modificaciones en la normativa para el sector acuicultor | Comienza en Olmué Encuentro Nacional de Directivos de Sernapesca 2013 | RT @SubpescaCL: Presidenta anunció que ya ingresó al Congreso Nacional el proyecto de regularización de caletas para pescadores artesanales |
| Sernapesca inmoviliza stock de Marine Harvest e inicia inspección de centro involucrado | Sernapesca investiga sospecha de virus ISA en Zona Norte de Aysén, cerca de Melinka | RT @meconomia: Ministro Céspedes pide aplicar todo el rigor de la ley para agresores de funcionarios de Sernapesca |
| Analisis de laboratorio permiten descartar uso de Cristal Violeta en salmones de Marine Harvest | Sernapesca publica Informe de Fiscalización de Actividades de Pesca y Acuicultura 2013 | Dir Reg @sernapesca en Primer Encuentro Regional de Mujeres de la #Pesca y #Acuicultura de @SernamChile #PuertoMontt |
| Sernapesca publicó su Cuenta Pública Participativa 2012. Consultas hasta el 06 de septiembre | Con uso de drones y GPS en embarcaciones se espera reforzar combate contra la pesca ilegal | RT @MaricelGonzalez: @sernapesca y Fondo Fomento para Pesca Artesanal potencia trabajo de pescadores con @LosReyesdelMar en #BahíaMansa |
| RT @comounpescado: Estudiantes de #quilpue en terreno conociendo #biodiversidad de las #algas @exploravalpo #lastorpederas #valparaiso | RT @AQUASocial: En marzo del próximo año terminará proceso 2015 de pago de #patente artesanal | RT @Chilepesca: Gremio de pescadores artesanales valora "evaluación" de #LeydePesca propuesta por Bachelet |
| RT @comounpescado: IFOP realiza taller sobre marea roja en regiones del sur-austral | RT @AQUASocial: Pesca artesanal del #Biobío permanece "en alerta" ante la creación de Ñuble Región | RT @risopulos: Mil 200 toneladas de harina de pesca ilegal incautó @sernapesca a empresas Marfood, Fiordo Austral, Landes y Blumar en la 8a |
| RT @biobio: Naciones Unidas felicita a Chile por reducir riesgos ante desastres naturales | En marzo de 2016 terminará proceso 2015 de pago de patente artesanal | Subse. @raulsunicog se reúne con dirigentes artesanales pelágicos del norte para analizar evaluación de la ley |
| RT @sernapescaIII: Por una #pescasustentable y un #consumoresponsable. Sáb 16 a las 12 en muelle de caldera @LosReyesdelMar | RT @AQUASocial: [#Marcha en #Valparaíso] Subsecretario Súnico se refirió a demandas de pescadores | Congreso aprueba proyecto de ley que establece normas permanentes para enfrentar catástrofes en el sector pesquero |

**Table A2.** Contents of the nine nearest tweets to cluster's centroid: March to May 2016.

| Cluster 1 | Cluster 2 |
|---|---|
| RT @COPASSurAustral: Biología de la microalga (Chattonelle spp.) que ha impactado a la industria del salmón @AQUASocial @ifop_periodista | el alga que está afectando a salmones no es dañiña para ser humano. Salmones afectados no irán a consumo humano de todas formas |
| RT @AQUASocial: La biología de la microalga que ha impactado a la industria del #salmón | Sernapesca extrema fiscalización ante llegada de mortalidad de salmones a plantas del Biobío |
| RT @CAPUERTOQUEMCHI: En #Chiloé solo consume mariscos certificados #marearoja #SemanaSanta | Monitoreo a navegación de wellboat que transporta mortandad de salmones ha zona de vertimiento a 120 kms de costa |
| @DefendamsChiloé @polycataldo @ribago Va estudio (inglés). Registro de muerte de peces, moluscos, invertebrados, etc | RT @mindefchile: ¿Viste @CNNChile? Masiva varazón de machas por #MareaRoja en #Chiloé |
| Informe de fiscalización de @sernapesca a vertimiento de mortandad de salmones | RT @AQUASocial: Bloom de algas: Presentan recurso para paralizar faenas de retiro de salmones muertos |
| RT @juancharlie37: @GreenpeaceCL Las catastrofes naturales o manipuladas hay que investigarlas con los que saben y no mienten ,bien por Greenpeace | Apareció @GreenpeaceCL la q no dice nada de la ley longueira, la q a mitad de discusión ley quedó muda, ???????????? |
| RT @FcoBoettcher: @AQUASocial @pymepes @PymeExporta @ProChile @rafasabat @Chilepesca @fetrapes excelente | La mano invisible de SalmonChile tras demandas de trabajadores en medio de la crisis de la industria |
| Informe de Subpesca da cuenta de delicado estado de conservación de los recursos pesqueros nacionales | RT @meconomia: Científicos exponen ante ministro en Comité Regional de emergencia en la Intendencia de Los Lagos por marea roja |
| RT @intendencialos1: Intendente de la Prida recorre buque en que viajan científicos que analizarán fenómeno de marea roja en #Chiloé | Inten. Ibáñez junto a pescadores artesanales hacen llamado a consumir pescados y mariscos de la Región. #Coquimbo |

**Table A3.** Contents of the nine nearest tweets to cluster's centroid (clusters 1, 2, and 3): June 2016 to January 2017.

| Cluster 1 | Cluster 2 | Cluster 3 |
|---|---|---|
| RT @Capuertovalp: A esta hora se realiza la tradicional romería en honor a San Pedro, patrono de los pescadores | Observador Científico Yerko Yutronich de IFOP asiste a curso en Francia | RT @SubpescaCL: ¡INFÓRMATE! Nuevos integrantes del Consejo de Investigación Pesquera y Acuicultura |
| Situación en Chile de mantarayas y tiburones es analizada por expertos en taller de trabajo | Acompañando a pescadores de Caleta Anahuac en la celebración de San Pedro y su 1era Cuenta Pública | RT @COREMagallanes: Ahora en Comisión Medio Ambiente: Sernapesca y Subpesca Magallanes informan acerca de Procesos de Fiscalización en Área |
| Logica de la gente del TPM, compro pesca ilegal, la transporto y la vendo. Si me multan la culpa es de @sernapesca | RT @SubpescaCL: Pasen el dato. Plazo hasta el 29 de julio para postular a los CCT Pesqueros | RT @SubpescaCL: ¡Buenos días y #FelizJueves! ?? Gobierno envía proyecto de ley para regular la captura de la reineta |
| RT @elcachapoal: IFOP realiza estudio sobre uso intraperitoneal de oxitetraciclina en salmones | RT @FAOpesca: ¿Qué son las islas de #basura de los #océanos y qué ocurre con el #plástico que tiramos? | RT @mop_chile: Avanza proyecto de una playa artificial y una caleta pesquera artesanal para el sector de La Chimba, en #Antofagasta |
| Fiscalización coordinada con Carabineros gracias a llamado a línea de atención de Sernapesca | Capacitación a agentes de aduanas en Concepción sobre sistema de exportación de Sernapesca #siscomex | RT @Corfo: #Coquimbo @meconomia #Corfo junto al sector pesquero artesanal en lanzamiento del programa |
| RT @lasotom: @radiolaclave Era obvio que la marea roja subiría su toxicidad con los desechos de salmoneras. Bien ahí @GreenpeaceCL | El @GobiernodeChile SOLO entregó el 20% de los bonos que le prometió a la comunidad de Chiloé. RECLAMA | Informe científico final ratificó que el vertimiento de salmones no tuvo relación con el fenómeno de Marea Roja |
| RT @muniteodoro: Histórica gestión entre Armada, Subpesca y Municipio de Teodoro Schmidt por E.C.M.P.O para comunidades Lafkenche | ¿Te enteraste? La Patagonia podría ser la próxima víctima de la saturación de la salmonicultura en Chile ?? | La extracción de recursos de la pesca artesanal representa en promedio el 32.7% del total del país. #TeQuieroCaleta |
| RT @sernapesca: Nueva normativa: Sernapesca reduce plazos para retiro de mortalidades masivas en centros de cultivo de salmónidos | RT @TodosPescamos: SalmonChile reconoce el drama humano que esconden las cifras de desempleo en la región | Junto a Gore #Maule l@s invitamos a participar de concurso para desarrollo productivo del sector pesquero artesanal |
| RT @minvu_loslagos: Vecinos y vecinas participando activamente en #CuentaPública. Sobre logros, sueños y trabajo en equipo se escucha en la | Apoyamos a pescadores artesanales para mejoramiento sanitario de #caletas en #OHiggins | Intendente, Seremi Salud y fiscalizadores de Salud y Sernapesca visitan Mercado Presidente Ibañez en #PtoMontt |

**Table A4.** Contents of the nine nearest tweets to cluster's centroid (clusters 4 and 5): June 2016 to January 2017.

| Cluster 4 | Cluster 5 |
| --- | --- |
| IFOP y SUBPESCA organizan panel sobre cambio climático @AQUASocial @raulsunicog @ucvradio @SomosNuestroMar @FISinfo | RT @meconomia: Min Céspedes participa en Comisión de Pesca de @CamaraDiputados que revisa proyecto que fortalece el @sernapesca |
| Sernapesca inicia investigación ante escape de salmones en Magallanes | Armada y @sernapesca decomisaron 516 kilos de centolla en #PuertoWilliams |
| RT @prensa_ptomontt: IFOP desarrolla estudio de jaibas en la zona centro-sur de Chile: El estudio se lleva a cabo entre las region | RT @gobercautin: En la #FeriaPinto @gobercautin y @sernapesca lanzan campaña de veda de la corvina. @josemontalva |
| RT @leo_paz_montes: 62% de avance presenta construcción infra. pesquera artesanal Caleta #Tongoy ejecutada x #ObrasPortuarias @mop_chile | RT @SubpescaCL: ¡ATENCIÓN #Chile! El pulpo del sur está en VEDA ???? desde #LosRíos a #Magallanes ¡Ayúdanos a protegerlo! #RespetaLaVeda |
| RT @sernapesca: Gran interés concita presentación del Sistema Integrado de Manejo de la Acuicultura, SIMA Austral | RT @intendencialos1: Ministro Economía, Intendente y parlamentarios anuncian extensión de aportes en localidades que sigan con marea roja |
| RT @DiarioOficialCL: @SubpescaCL Aprueba Plan de Manejo para la Pesquería de Sardina común y Anchoveta, V a la X Región | RT @intendencialos1: Intendente, Seremi Salud y fiscalizadores de Salud y Sernapesca visitan Mercado Presidente Ibañez en #PtoMontt |
| RT @sernapesca: Sólo 10 citaciones por no respetar la veda de la Merluza común en sept. Notable compromiso de pescadores | Entregamos apoyo social y estudios técnicos a trabajadores y ex trabajadores del sector pesquero industrial |
| RT @AQUASocial: Armada probará nuevo sistema de vigilancia contra la pesca ilegal | La corvina SIGUE en veda en todo #Chile Ayúdanos a protegerla y denuncia toda extracción ilegal a @sernapesca |
| RT @seremisalud10: Se acerca #SemanaSanta y la Mesa Marea Roja #Quellón se prepara para asegurar que los mariscos que se vendan cumplan | Ministro de Economía junto a @SubpescaCL reciben el informe final de la Comisión Científica Marea Roja |

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
