# Peer review of "A Twitter-Lived Red Tide Crisis on Chiloé Island, Chile: What Can Be Obtained for Social-Ecological Research through Social Media Analysis?"

_sustainability, doi:10.3390/su12208506_

Round 1

Reviewer 1 Report

The article makes an interesting contribution about the usefulness of social media analysis, specifically Twitter, to identify in social-ecological crises the general dynamics of behavior and communication of the actors involved. This approach can complement other qualitative and quantitative studies on social-ecological crises, in order to understand in greater depth the relationships between subjects and systems in the making of meaning, in the search for causes, the identification of conflicts and solutions to this kind of crisis.

The background section of the article is complete and clarifying to situate a reader who is not familiar with the context in which the 2016 Chiloe red tide occurred. The review of the literature on social media analysis as well as the 2016 crisis in Chiloe is complete, clarifying and updated.

The research design is correct and well explained. For this section I have only two minor questions: Why is the Google Trends data collected from 2004 to February 1st 2020? Why is the data collected from GT, not from the same periods as the Twitter analysis (2013 to January 2017)? I suggest you could include a brief explanation in subsection 3.1 (Line 176)

The results are clearly presented, but they are very extensive, and it is difficult to follow the relationship between the different evidence presented. The tables and figures help to understand them. But, in order to make the results much clearer and useful for the scientific community, the affected community and society in general, please: include in section "6. Conclusion" some of the most meaningful examples of the relevant dynamics identified in the social media analysis, which contribute to the collective learning process of the affected community and their governments so that they can act in a more coordinated and sustainable way in future crises.

Author Response

Response to Reviewer 1 Comments

Point 1: The article makes an interesting contribution about the usefulness of social media analysis, specifically Twitter, to identify in social-ecological crises the general dynamics of behavior and communication of the actors involved. This approach can complement other qualitative and quantitative studies on social-ecological crises, in order to understand in greater depth, the relationships between subjects and systems in the making of meaning, in the search for causes, the identification of conflicts and solutions to this kind of crisis. The background section of the article is complete and clarifying to situate a reader who is not familiar with the context in which the 2016 Chiloe red tide occurred. The review of the literature on social media analysis as well as the 2016 crisis in Chiloe is complete, clarifying and updated.

The research design is correct and well explained. For this section I have only two minor questions: Why is the Google Trends data collected from 2004 to February 1st 2020? Why is the data collected from GT, not from the same periods as the Twitter analysis (2013 to January 2017)? I suggest you could include a brief explanation in subsection 3.1 (Line 176).

Response 1: Thank you for the indication. We have changed the Figures. Now Figures 2 and 3 show data from the same period as the Twitter analysis. There was indeed no real need for previous data. See L308.

Point 2: The results are clearly presented, but they are very extensive, and it is difficult to follow the relationship between the different evidence presented. The tables and figures help to understand them. But, in order to make the results much clearer and useful for the scientific community, the affected community and society in general, please: include in section "6. Conclusion" some of the most meaningful examples of the relevant dynamics identified in the social media analysis, which contribute to the collective learning process of the affected community and their governments so that they can act in a more coordinated and sustainable way in future crises.

Response 2: We have included several paragraphs with our conclusions, particularly stressing the point of learning processes. See L726-797:

Several conclusions can be drawn from our analysis. We describe them in detail.

First, a fundamental resource for constructing political alliances, learning processes and reflexiveness in social-ecological crises is to share a common source of meaning-making for different actors. In our case study, this common source is the semantic constellation of ‘fishing’ with all its variances (artisanal fishing, industrial fishing, illegal fishing, trawl fishing, ban on fishing, fishing law, fishermen). This constellation connects different and even opposite views and interests, yet it turns into a common ground for discussing and acting within a shared semantic framework that facilitates collective action for and against. While there might be conflicts of goals and concerns, they are processed within the semantic constellation of fishing.

Second, that actors share a semantic constellation does not mean, however, that conflicting interests and worldviews cannot be expressed. Social diversity lies at the basis of social-ecological systems [17]. There is, for example, contradictory performative effects between fishing and ban on fishing, or between artisanal fishing and industrial fishing. These rather opposite positions lead to political alliances (such as those between fishermen and NGOs criticizing the salmon industry), generalized conflicts (between technical agencies and the government, on the one side, and fishermen and NGOs, on the other), and partial consensus (between fishermen and the government) on the causes and development of the crisis as well as on the measures supporting affected fishermen. To that extent, we confirm previous findings in the literature regarding the competing interpretations in crisis processes.

Third, the force of social diversity emerges not only from the different structural positions that the involved actors occupy, but also from the dissimilar historical experience actors have regarding prior critical situations. Our results show (see particularly Figure 16) that fishermen do not communicate as a homogeneous actor. They are divided into two groups: while one group shares discursive contents with other actors, the other one has a discourse of its own. Drawing on previous research, we attribute this divide to different political stances on Chiloé Island: the former demanding redistribution of costs and benefits of modernization, and the latter emphasizing the uniqueness of Chiloé’s identity [44]. This is a rather regular dynamics in accelerated modernization processes in which local groups follow different paths. In these cases, the divide is an obstacle that has to be overcome by collective action and learning processes [20, 115].

Fourth, besides the competing interpretations in crisis processes, the literature stresses the breakdown of accustomed structures of meaning at the initial stages of the crisis. In our findings, there is no evidence supporting this. Affected actors begin with the reflexive work of making sense of the crises right after the major algal bloom. They look for the causes, speculate on who is responsible for the disaster, publicly accuse involved actors, organize themselves and demand solutions. It is truth that there is a regime shift [136, 137] in the main topic of communication after the outbreak of the red tide crisis (from ‘fishing law’ to ‘red tide’ and ‘fishermen’), but there is no breakdown in meaning-making. In fact, communication is supported by the semantic constellation of 'fishing', i.e. it remains communicating on topics that are relevant for the fundamental source of meaning-making of island inhabitants. This quick response shows the reflexiveness of local actors, particularly fishermen and NGOs, who immediately engage in making sense of the new situation, materializing it into political actions and triggering learning processes.

Fifth, continuities and discontinuities in meaning-making are a relevant factor that learning processes have to consider. Collective learning from social-ecological crises does not mean that every actor has to agree with the one and only interpretation of the crisis. This would involve an important loss in social diversity. Rather, collective learning means increasing the capacity of self-organization for resilience and sustainable development particularly in local communities and vulnerable areas [17-19]. With the 2016 red tide crisis in Chiloé, the local community has learnt (or has confirmed previous learning) that political alliances are useful for bringing local problems to a public forum, they have developed the experience of negotiating with the government, and they have assembled a memory of the crisis through locally based social organizations that works as a political and normative repository for future events. For example, in the current COVID-19 pandemic, Chiloé Island is one of the least affected territories in Chile in terms of the spread of the virus, with 44 out of 14,139 active cases nationwide as of October 8, 2020 [138]. In co-ordination with local authorities, the community has self-organized to control access points to the island and to inform the inhabitants on the preventive measures. The experience with recent crises seems to work in the current critical situation. Further research can address this point.

Finally, research on the 2016 red tide crisis on Chiloé Island, the most significant algal bloom in the southern hemisphere in the last century, has been mainly conducted through qualitative methods (in-depth interviews, framing methods). In this article, we applied a multi-method approach to Twitter data with which we have supplemented and discussed existing findings in the literature and produced new insights on the 2016 red tide crisis on Chiloé Island, Chile. Our article supports the argument, brought forward by multidisciplinary scholarship in resilience research, disaster risk reduction, neogeography and social-ecological systems, about the need of developing and field-testing mixed methods that connect semantic analysis with the new opportunities offered by social media, voluntary geographic information and other data sources. Taking advantage of the new opportunities increasingly made available by current innovations in information technology is crucial for improving our understanding of social-ecological systems. Accomplishing this depends on the research design (different actors’ views distributed in a long-time span) and on the combined character of the techniques applied. Good qualitative research can be thus supplemented with new possibilities of capturing meanings from the past that cannot be retrieved otherwise. This is particularly relevant for studying social-ecological crises and supporting collective learning processes that point out towards increased resilience capacities and more sustainable trajectories in affected communities.

Reviewer 2 Report

This paper presents an interesting approach to ecological crises and disasters through the use of social media. To enrich and strengthen the theoretical framework of the paper, I would recommend referring to neogeography and voluntary geographical information which are the fields of study that deal with these issues. In addition, some reference to the literature on Disaster Risk Reduction on social media and the use of Twitter in the management phase of communication processes by governance would also be appropriate. It would also be interesting a comparison with the current pandemic situation to be included in the (rather short) conclusions. Have social media affected communication in this pandemic crisis management?

Author Response

Response to Reviewer 2 Comments

Point 1: This paper presents an interesting approach to ecological crises and disasters through the use of social media. To enrich and strengthen the theoretical framework of the paper, I would recommend referring to neogeography and voluntary geographical information which are the fields of study that deal with these issues.

Response 1: In the Introduction, we have included a paragraph including relevant literature on neogeography, volunteered geographical information, and participatory geographic information systems. See L57-65:

Additionally, this effort can contribute to the emerging scholarship in the realm of neogeography, volunteered geographical information, and participatory geographic information systems. This scholarship signals the opportunities associated to crowd-sourced information and other sources of big data for the analysis of a variety of social phenomena [26], including crises [27,28], environmental degradation [29], inequalities [30], and disaster consequences and response [31–33]. Neogeography has also highlighted how the growing availability and accessibility of user-generated information transforms the comprehension of actors, practices and contents of socio-geographical analysis and place-making [34,35], thereby opening new avenues for citizen engagement in knowledge production [36,37] and generating new challenges in terms of quality assurance [38,39] and critical analysis [40].

Point 2: In addition, some reference to the literature on Disaster Risk Reduction on social media and the use of Twitter in the management phase of communication processes by governance would also be appropriate.

Response 2: We have introduced a paragraph at L100-116 addressing this point:

A third strand of literature discusses the potential of Twitter and other social media as sources of information to reduce risks, identify and prevent crisis situations and put in place early warning systems. This scholarship debates different advanced methods and techniques for Twitter mining, labelling and automated screening, as well as the potential usefulness of crossing social media data with other information sources such as geospatial data [65–70]. This literature shows the relevance of Twitter information during disasters so that authorities can make better decisions [71] and also highlights Twitter importance to create awareness and minimize possibles damages [72–74].

Point 3: It would also be interesting a comparison with the current pandemic situation to be included in the (rather short) conclusions. Have social media affected communication in this pandemic crisis management?

Response 3: This is an interesting point for further research. Nonetheless, we have included a reference to this topic in the Conclusions at the end of the paragraph at L767-781:

Fifth, continuities and discontinuities in meaning-making are a relevant factor that learning processes have to consider. Collective learning from social-ecological crises does not mean that every actor has to agree with the one and only interpretation of the crisis. This would involve an important loss in social diversity. Rather, collective learning means increasing the capacity of self-organization for resilience and sustainable development particularly in local communities and vulnerable areas [17–19]. With the 2016 red tide crisis in Chiloé, the local community has learned (or has confirmed previous learning) that political alliances are useful for bringing local problems to a public forum, they have developed the experience of negotiating with the government, and they have assembled a memory of the crisis through locally based social organizations that works as a political and normative repository for future events. For example, in the current COVID-19 pandemic, Chiloé Island is one of the least affected territories in Chile in terms of the spread of the virus, with 44 out of 14,139 active cases nationwide as of October 8, 2020 [138]. In co-ordination with local authorities, the community has self-organized to control access points to the island and to inform the inhabitants on the preventive measures. The experience with recent crises seems to work in the current critical situation. Further research can address this point.

Reviewer 3 Report

REVIEW OF THE MANUSCRIPT: SUSTAINABILITY - 943467

The manuscript is within the aim and scope of this Journal. The theme the authors aim to discuss is of relevance for sustainability science and social-ecological research. The paper is well structured, the research design is appropriate, the methods have been adequately described, and the research findings adequately presented and discussed. However, it would be very desirable if the authors could engage a bit more with the international audience, especially with the international academic community working on social-ecological systems and conducting social-ecological research. At this regard I would suggest to enhance the introduction and conclusion with further and updated references to SES literature.

More precisely, the reflection, which the authors provided only at the discussion section, about learning processes that could be detected through social media and Twitter analysis, should be made earlier in the paper. For social-ecological research, identifying the cognitive and interactional processes that enable social learning and ‘transformation towards sustainability’ (i.e. addressing vulnerabilities, risks and impacts) in SES crisis conditions and, thus, the human agency within social-ecological systems, is crucial.

An interesting stream of research in SES studies remarked that “the emphasis on human relations with the environment has led to a weak theorization of the “social” in the SES model” (Fabinyi et al 2014, online) and it advocated for an enhanced social-ecological understanding of social learning and transformation dynamics which should better deepen social theories about human agency, including human intentionality, social learning and the underpinning human feelings, attitudes and behaviours in SES crisis conditions, and concepts such as equity, power relationships and social change and deliberativeness (Robards et al., 2011; Armitage et al., 2012; Cote and Nightingale, 2012; Wilson, 2013; Fabinyi et al., 2014; Walsh-Dilley et al., 2016; Imperiale and Vanclay, 2016a, 2016b). In their work, Experiencing Community Resilience in action: learning from post-disaster communities, Imperiale and Vanclay (2016), for example, used an ethnographic approach to deepen the human agency in SES crisis conditions. Also, adopting a SES approach to resilience, they made a quite interesting literature review on SES approaches to human agency, learning, transformation and resilience, the authors can draw on.

Overall, the paper is almost ready for publication, however, I suggest minor revision to let the authors better stress the gaps in SES literature and research, specifically on human agency, and engage with the international research strand in the SES field that advocated for more focussed investigation to fill this gap and conceptualise  learning and transformation processes at the local community level (i.e. community resilience) and at all levels of social-ecological governan ce (i.e. social resilience). By doing so, the authors will better underline the relevance of social media analysis and the potential contribution Twitter analysis can have to enhance SES research on human agency and on social learning and transformation towards sustainability in society, in times of crises and disasters.

Author Response

Response to Reviewer 3 Comments

Point 1: The manuscript is within the aim and scope of this Journal. The theme the authors aim to discuss is of relevance for sustainability science and social-ecological research. The paper is well structured, the research design is appropriate, the methods have been adequately described, and the research findings adequately presented and discussed. However, it would be very desirable if the authors could engage a bit more with the international audience, especially with the international academic community working on social-ecological systems and conducting social-ecological research.

At this regard I would suggest to enhance the introduction and conclusion with further and updated references to SES literature. More precisely, the reflection, which the authors provided only at the discussion section, about learning processes that could be detected through social media and Twitter analysis, should be made earlier in the paper. For social-ecological research, identifying the cognitive and interactional processes that enable social learning and ‘transformation towards sustainability’ (i.e. addressing vulnerabilities, risks and impacts) in SES crisis conditions and, thus, the human agency within social-ecological systems, is crucial. An interesting stream of research in SES studies remarked that “the emphasis on human relations with the environment has led to a weak theorization of the “social” in the SES model” (Fabinyi et al 2014, online) and it advocated for an enhanced social-ecological understanding of social learning and transformation dynamics which should better deepen social theories about human agency, including human intentionality, social learning and the underpinning human feelings, attitudes and behaviour in SES crisis conditions, and concepts such as equity, power relationships and social change and deliberativeness (Robards et al., 2011; Armitage et al., 2012; Cote and Nightingale, 2012; Wilson, 2013; Fabinyi et al., 2014; Walsh-Dilley et al., 2016; Imperiale and Vanclay, 2016a, 2016b). In their work, Experiencing Community Resilience in action: learning from post-disaster communities, Imperiale and Vanclay (2016), for example, used an ethnographic approach to deepen the human agency in SES crisis conditions. Also, adopting a SES approach to resilience, they made a quite interesting literature review on SES approaches to human agency, learning, transformation and resilience, the authors can draw on.

Response 1: Many thanks for the references. We have introduced new paragraphs at L43-56 and L152-178:

It is, thus, our contention that these methods can associate the structural approach to social-ecological crises –by identifying behavioral patterns and complexly interrelated semantic networks– with the more phenomenological actor-centered approach –by recognizing shared meanings, common experiences, and learning processes oriented to reinforce community resilience and the transformation towards sustainability coming from different social actors. This attempt of constructing a mixed approach to social-ecological systems (SES) by addressing structural realities and actors' experiences through social media analysis can contribute to the discussions on the role of the social in SES literature. Most of the human behaviour having relevant impacts on the environment neither comes from isolated individual decisions nor from their simple aggregation, but from socially constructed constellations of meaning wherein values, knowledge, social diversity and power relations play a fundamental role. Observing how these meaningful constellations work and how they motivate resilience practices and learning processes in crisis situations is thus crucial for a better understanding of the social-ecological nexus [17-25].

And at L152-178:

In this vein, social-ecological systems (SES) literature is a fertile field of research. The SES approach investigates the complex adaptive system dynamics linking human and natural systems [101], explores their trajectories and interdependencies at multiple scales [102,103] as well as pathways for the sustainable governance of these systems [104,105]. In recent years, increasing attention has been devoted to the role played in social-ecological systems sustainability by mismatches of scale between social-ecological phenomena and institutional arrangements [106], time-delayed ecological feedbacks on biodiversity upon human activities [107], the relevance of values, knowledge, social diversity and power relations in the processes of meaning-making leading to resilience practices, learning, and sustainability [17–24], and the complex interactions between interlocked social-ecological systems [108], events which may have been playing a salient role in the 2016 Chiloé red tide crisis.

Fisheries and fishing ecosystems, in particular, have been abundantly explored within social-ecological research. Fishing resources are a classical example of a common-pool resource [109], and thus prone to over-exploitation and under-caring. This has led to ample interest in exploring the possibility of setting up sustainable fishing communities, particularly through the 166 use of polycentric frameworks and principles [110,111], and has also led to investigate the resilience of fishing communities in the face of internal and external stresses [112–114].

Within this context, Chiloé is a widely employed case study on the sustainability of fishing ecosystems, especially in association with the sustainability of salmon and shellfish aquaculture and their environmental and social impacts on the island and its population [115–119]. The 2016 events have contributed to boosting this literature, with new contributions multiplying in recent years, covering long-term socio-cultural and social-ecological trends and transformations contributing to explain the crisis [44,120,121], the specific influence played by the salmon dumping on the crisis [46];local perceptions on the crisis and its relationship with climate and environmental change [122] and the different frames employed in media coverage of the crisis [123,124]. Noticeably, only one of these studies [125] makes use of social media analyses, and as we observe in detail in the next section, it is focused on the limited use of these platforms for concrete political actions than on the evolution of social media communications as a way to understand the underlying dynamics of the crisis.

Point 2: Overall, the paper is almost ready for publication, however, I suggest minor revision to let the authors better stress the gaps in SES literature and research, specifically on human agency, and engage with the international research strand in the SES field that advocated for more focused investigation to fill this gap and conceptualise learning and transformation processes at the local community level (i.e. community resilience) and at all levels of social-ecological governance (i.e. social resilience). By doing so, the authors will better underline the relevance of social media analysis and the potential contribution Twitter analysis can have to enhance SES research on human agency and on social learning and transformation towards sustainability in society, in times of crises and disasters.

Response 2: In addition to the inclusion of relevant SES literature in the Introduction and into the Literature review section, we have addressed this general concern in the new Conclusions. See new paragraphs at L726-797:

Several conclusions can be drawn from our analysis. We describe them in detail.

First, a fundamental resource for constructing political alliances, learning processes and reflexiveness in social-ecological crises is to share a common source of meaning-making for different actors. In our case study, this common source is the semantic constellation of ‘fishing’ with all its variances (artisanal fishing, industrial fishing, illegal fishing, trawl fishing, ban on fishing, fishing law, fishermen). This constellation connects different and even opposite views and interests, yet it turns into a common ground for discussing and acting within a shared semantic framework that facilitates collective action for and against. While there might be conflicts of goals and concerns, they are processed within the semantic constellation of fishing.

Second, that actors share a semantic constellation does not mean, however, that conflicting interests and worldviews cannot be expressed. Social diversity lies at the basis of social-ecological systems [17]. There is, for example, contradictory performative effects between fishing and ban on fishing, or between artisanal fishing and industrial fishing. These rather opposite positions lead to political alliances (such as those between fishermen and NGOs criticizing the salmon industry), generalized conflicts (between technical agencies and the government, on the one side, and fishermen and NGOs, on the other), and partial consensus (between fishermen and the government) on the causes and development of the crisis as well as on the measures supporting affected fishermen. To that extent, we confirm previous findings in the literature regarding the competing interpretations in crisis processes.

Third, the force of social diversity emerges not only from the different structural positions that the involved actors occupy, but also from the dissimilar historical experience actors have regarding prior critical situations. Our results show (see particularly Figure 16) that fishermen do not communicate as a homogeneous actor. They are divided into two groups: while one group shares discursive contents with other actors, the other one has a discourse of its own. Drawing on previous research, we attribute this divide to different political stances on Chiloé Island: the former demanding redistribution of costs and benefits of modernization, and the latter emphasizing the uniqueness of Chiloé’s identity [44]. This is a rather regular dynamics in accelerated modernization processes in which local groups follow different paths. In these cases, the divide is an obstacle that has to be overcome by collective action and learning processes [20, 115].

Fourth, besides the competing interpretations in crisis processes, the literature stresses the breakdown of accustomed structures of meaning at the initial stages of the crisis. In our findings, there is no evidence supporting this. Affected actors begin with the reflexive work of making sense of the crises right after the major algal bloom. They look for the causes, speculate on who is responsible for the disaster, publicly accuse involved actors, organize themselves and demand solutions. It is truth that there is a regime shift [136, 137] in the main topic of communication after the outbreak of the red tide crisis (from ‘fishing law’ to ‘red tide’ and ‘fishermen’), but there is no breakdown in meaning-making. In fact, communication is supported by the semantic constellation of 'fishing', i.e. it remains communicating on topics that are relevant for the fundamental source of meaning-making of island inhabitants. This quick response shows the reflexiveness of local actors, particularly fishermen and NGOs, who immediately engage in making sense of the new situation, materializing it into political actions and triggering learning processes.

Fifth, continuities and discontinuities in meaning-making are a relevant factor that learning processes have to consider. Collective learning from social-ecological crises does not mean that every actor has to agree with the one and only interpretation of the crisis. This would involve an important loss in social diversity. Rather, collective learning means increasing the capacity of self-organization for resilience and sustainable development particularly in local communities and vulnerable areas [17-19]. With the 2016 red tide crisis in Chiloé, the local community has learnt (or has confirmed previous learning) that political alliances are useful for bringing local problems to a public forum, they have developed the experience of negotiating with the government, and they have assembled a memory of the crisis through locally based social organizations that works as a political and normative repository for future events. For example, in the current COVID-19 pandemic, Chiloé Island is one of the least affected territories in Chile in terms of the spread of the virus, with 44 out of 14,139 active cases nationwide as of October 8, 2020 [138]. In co-ordination with local authorities, the community has self-organized to control access points to the island and to inform the inhabitants on the preventive measures. The experience with recent crises seems to work in the current critical situation. Further research can address this point.

Finally, research on the 2016 red tide crisis on Chiloé Island, the most significant algal bloom in the southern hemisphere in the last century, has been mainly conducted through qualitative methods (in-depth interviews, framing methods). In this article, we applied a multi-method approach to Twitter data with which we have supplemented and discussed existing findings in the literature and produced new insights on the 2016 red tide crisis on Chiloé Island, Chile. Our article supports the argument, brought forward by multidisciplinary scholarship in resilience research, disaster risk reduction, neogeography and social-ecological systems, about the need of developing and field-testing mixed methods that connect semantic analysis with the new opportunities offered by social media, voluntary geographic information and other data sources. Taking advantage of the new opportunities increasingly made available by current innovations in information technology is crucial for improving our understanding of social-ecological systems. Accomplishing this depends on the research design (different actors’ views distributed in a long-time span) and on the combined character of the techniques applied. Good qualitative research can be thus supplemented with new possibilities of capturing meanings from the past that cannot be retrieved otherwise. This is particularly relevant for studying social-ecological crises and supporting collective learning processes that point out towards increased resilience capacities and more sustainable trajectories in affected communities.